# State2Explanation: Concept-Based Explanations to Benefit Agent Learning and User Understanding

**Devleena Das**
School of Interactive Computing
Georgia Institute of Technology
ddas41@gatech.edu

**Sonia Chernova**
School of Interactive Computing
Georgia Institute of Technology
chernova@gatech.edu

**Been Kim**
Google Research
beenkim@google.com

## Abstract

As more non-AI experts use complex AI systems for daily tasks, there has been an increasing effort to develop methods that produce explanations of AI decision making that are understandable by non-AI experts. Towards this effort, leveraging higher-level concepts and producing concept-based explanations have become a popular method. Most concept-based explanations have been developed for classification techniques, and we posit that the few existing methods for sequential decision making are limited in scope. In this work, we first contribute a desiderata for defining "concepts" in sequential decision making settings. Additionally, inspired by the Protégé Effect which states explaining knowledge often reinforces one's self-learning, we explore how concept-based explanations of an RL agent's decision making can in turn improve the agent's learning rate, as well as improve end-user understanding of the agent's decision making. To this end, we contribute a unified framework, State2Explanation (S2E), that involves learning a joint embedding model between state-action pairs and concept-based explanations, and leveraging such learned model to both (1) inform reward shaping during an agent's training, and (2) provide explanations to end-users at deployment for improved task performance. Our experimental validations, in Connect 4 and Lunar Lander, demonstrate the success of S2E in providing a dual-benefit, successfully informing reward shaping and improving agent learning rate, as well as significantly improving end user task performance at deployment time.

## 1 Introduction

Black-box AI systems are increasingly being deployed to help end-users with everyday tasks. Examples include doctors leveraging decision support systems to aid in diagnosis [39], warehouse managers relying on robots for goods transportation [8], and drivers using autonomous vehicles for assisted driving [18]. To increase the transparency of these black-box models, researchers have developed numerous techniques to provide explanations of agent decision making [45, 2, 23, 51, 14, 9, 30, 15].

A popular method towards non-expert friendly explanations has been to attribute higher-level "concepts" to an agent's decision making, and these concepts have primarily been used to explain classification-based AI systems [30, 33, 54, 19]. An example of a concept-based explanation for a classification model includes, "wing color" for a bird class label [19]. In sequential decision making, concept-based explanations have been less widely explored, and existing works leverage preconditions of states and action costs [48] or logical formulas [22] as "concepts".

37th Conference on Neural Information Processing Systems (NeurIPS 2023).

In this work, we posit that concept-based explanations may not *only* benefit the end-user but also benefit the AI agent. Our claim is loosely motivated by the Psychological phenomenon, the Protégé Effect, that states explaining material to another student also helps the explainer learn and reinforce material [7, 24, 17]. Thus the objective of our work is to demonstrate the dual-benefit of concept-based explanations to both the end-user for improved understanding, as well as the AI agent for improved learning rate. Note, prior works, typically in Reinforcement Learning (RL), have explored various methods to improve agent learning, a common approach including the use of human feedback and natural language commands as a method of reward shaping to improve RL agent sample efficiency and learning rate [38, 50, 20]. However, these methods have focused on leveraging language as a one-way benefit to the agent, not considering how language provided to the agent during training may also benefit end-user understanding at deployment. Therefore in our work, we explore the following research question: *Can concept-based explanations have a two-way benefit to both the user and agent, such as to improve end-user understanding of a deployed agent's decision making, while also improving the agent's learning rate via reward shaping?"* As a solution, we contribute a unified framework, **State2Explanation (S2E)**, that learns a single joint embedding model between agent states and associated concept-based explanations and leverages such learned model to (1) inform reward shaping for agent learning, and (2) improve end-user understanding of the agent's decision making at deployment.

Additionally, we contribute a desiderata of what entails a "concept" in a concept-based explanation for sequential decision making systems. Given that sequential decision making agents engage in long-term interaction with their environment, we posit that the scope of concept-based explanations should span *beyond* representing preconditions and action cost [48], and control logic [22]. Specifically, we believe concept-based explanations in sequential decision making should function at a much higher-level of abstraction, highlighting knowledge that are applicable across multiple states, and most importantly, expressing a positive or negative influence towards the agent's goal. Our claim is motivated by the importance of the agent's goal in a sequential decision making formulation [44, 49].

**Contributions**. (1) We provide desiderata for what constitutes a "concept" in concept-based explanations of sequential decision making. (2) We introduce a novel framework, State2Explanation (S2E) which learns a joint embedding model between agent states-action pairs and concept-based explanations to provide a benefit to both the agent, by informing reward shaping, as well as the end user by providing explanations that improve user task performance. We perform both model and user evaluations of our S2E framework in two complex RL domains, Connect 4 and Lunar Lander.

Our model evaluations demonstrate that S2E can successfully inform reward shaping, comparable to expert-defined reward functions. Additionally, our user evaluations demonstrate that S2E can successfully provide meaningful concept-based explanations to end users such that exposure to our explanations significantly improve user task performance.

## 2 Related Work

In this work, we explore the utility of concept-based explanations in providing a two way benefit to both the AI agent during training, via reward shaping, as well as end user at deployment time, via improved user task performance. Below, we highlight related prior work.

**Reward Shaping using Language**   Many prior works have focused on improving an agent's learning rate or sample efficiency [35, 21]. Closely related to our work are methods providing reward shaping via language. Specifically, [38] propose ELLA, which leverages a relevance classifier to associate a lower level instruction to a higher level task for providing a subtask shaping reward. In [20], the authors leverage natural language instruction to derive potential-based shaping rewards that encourage an RL agent to more frequently select relevant actions in the game of Montezuma's Revenge. Similarly, [50] learn a joint embedding model between subgoal language commands and agent states to provide shaping rewards for completing subtasks in StarCraft II. Specifically, [50] demonstrate that their model can generalize to unseen, semantically similar language commands, and improve agent learning rate. In our work, we take inspiration from [50], and learn a joint embedding model between agent states and associated natural language commands to inform agent reward shaping. However, our work differs in that our natural language commands are *concept-based explanations* of the associated state, and not subgoal descriptions required for task completion. By learning a common embedding space between state explanations and state representations, our work

provides a *two-way* benefit to both the RL agent for learning as well as end-user for improved understanding.

**Explanations for non-AI Experts** Prior works have established that concept-based explanations for classification tasks provide explanations that are meaningful to end-users [30, 33, 54, 19]. Methods for concept-based explanations have included Concept Activation Vectors [30], Concept Bottleneck Models [33], and Concept Embedding Models [54]. For classification tasks, "concepts" have primarily represented human-interpretable features that a model's prediction is sensitive to (i.e. 'wing color' for image classification.) [33]. Related to sequential decision making, [26] provide concept-based explanations of 3D action recognition CovNets by clustering learned, human-interpretable features (i.e. "grass" or "hand") from segmented videos. In [48], concept-based explanations for sequential decision making are formulated in relation to state preconditions (e.g. "the action *move-left* failed in the state as the precondition *skull-not-on-left* was false in the state") and action costs (e.g., "executing the action *attack* in the presence of the concept *skull-on-left*, will cost at least *500*"). Specifically, in [48], concepts have represented any factual statement a user associates to a given state, such as "skull-on-left" in the previous example. Similarly, in [22], concepts have represented logical formulas for summarizing RL policies. Additionally, [29] propose a prototype wrapper network for training interpretable RL policies with user-defined prototypes. Similar to [48], the authors in [29] define prototypes as factual observations about a state, such as "turn right" or "turn left". Other explanation generation methods have demonstrated that end user understanding is improved when explanations are presented in natural language rather than features or labels [14, 5, 9, 10]. Inspired by these findings, our work contributes the first method for producing natural language concept-based explanations, where we define a general desiderata of concepts that goes beyond action costs and preconditions to capture goal-driven decision making.

## 3 Concept Based Explanations for Sequential Decision Making

We posit that the objective of a concept-based explanation, in sequential decision making settings, is to communicate higher-level knowledge that aids the agent and/or user in reasoning about the utility of taking a given action from a specific state. In this section, we provide a general definition of concept-based explanations and "concepts", in sequential decision making, that extends beyond prior usage of state preconditions and action-costs [48], and control logic [22]. Specifically, we define a concept-based explanation, $\mathcal{E}_{C_i}^i$, as one that presents, in natural language form, the set of concepts, $C_i = \{c_j..c_m\}$, that can describe a particular state-action pair $(s_i, a_i)$ in the context of goal $G$. This naturally poses the question of: *"What is a concept $c_j$?"* Below are several desiderata for defining a concept in sequential decision making settings:

1. **A concept should be grounded in human domain knowledge**. A concept, $c_j$, should be contextualized in the overall task domain. For example, although an agent's goal may be to optimize its Q-function, we posit that "higher Q-value" is not a valid concept. Instead, a concept should represent a higher-level abstraction of what the "higher Q-value" relates to in the domain. For example, in Chess, an agent receiving a "higher Q-value" for a particular state-action pair may be translated by a domain expert to "capturing a queen".

2. **A concept should relate to the task goal**. A concept, $c_j$, is an abstracted representation of $(s_i, a_i)$ that encompasses how $(s_i, a_i)$ may lead to or inhibit the agent's goal $G$. This claim is motivated by the general objective of a sequential decision making agent, that is to perform a sequence of actions $\langle a_1, a_2, ..., a_n \rangle, a_i \in A$, defined via a plan or policy $\pi$, that transforms the agent's current state $I$ to its goal state $G$ [44, 49]. For example, we do not consider the description "Knight in A4" as a valid concept, since, by itself, it does not provide relation to the game's objective. However, we would consider "king safety" as a valid concept, since it relates to game's objective of checkmate.

3. **A concept should be generalizable.** A concept, $c_j$, should generalize across multiple $(s_i, a_i)$ pairs. This means that a concept is typically *not* a unique description of a single $(s_i, a_i)$. For example, a valid concept in Chess is "checkmate" since the abstraction of "checkmate" can generalize to many different $(s_i, a_i)$. An invalid concept involves detailing the entire game board by piece position.

Note, above we define a concept, $c_j \in C_i$, in the context of describing a single state-action pair, $(s_i, a_i)$. However, in many domains a meaningful concept $c_j$ can only be defined in terms of a

sequence of state-action pairs. This is especially true in domains in which state-action pairs are sampled at high-frequency, such as robotics and video games [40, 56]. For example, concepts such as "rotate wrist" or "bend arm" can only be defined by a series of state-action pairs, as opposed to a single state-action pair. As a result, we define concept-based explanation in these domains as $\mathcal{E}_{C_i}^{[i,i+n]}$, which presents the set of concepts $C_i = \{c_j..c_m\}$, for a series of $(s_{[i,i+n]}, a_{[i,i+n]})$ pairs.

Finally, we highlight that some $(s_i, a_i)$ or $(s_{[i,i+n]}, a_{[i,i+n]})$ pairs may not have a concept-based explanation associated with them. In other words, $C_i = \emptyset$ for some $(s_i, a_i)$ pair or $(s_{[i,i+n]}, a_{[i,i+n]})$ pairs. This is especially true in long horizon tasks, in which a $(s_i, a_i)$ may not relate directly to $G$ but is important for another, future $(s_{i+n}, a_{i+n})$ pair which then leads to $G$. For example, "Move the pawn to D5 so that two moves from now you may capture the rook". Our definition of concept-based explanations do not consider these cases, and is an opportunity for future work.

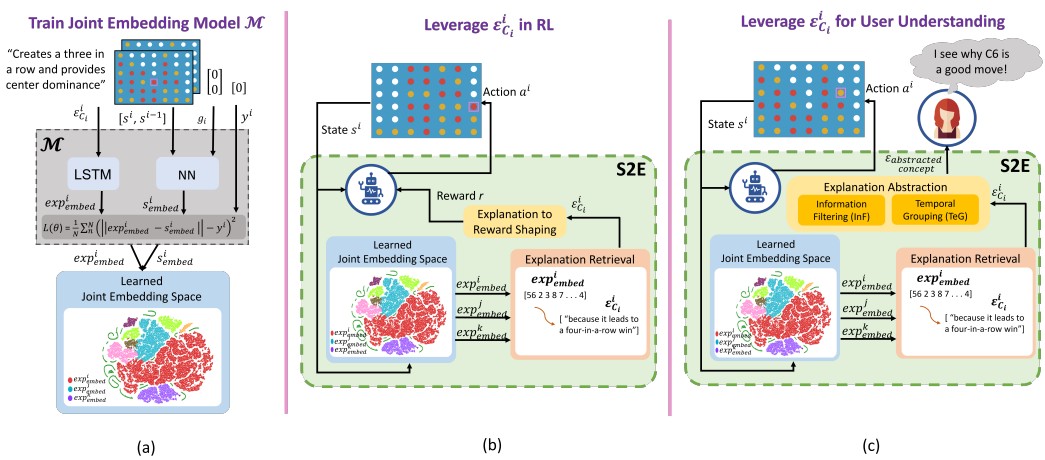

Figure 1: Our S2E framework involves (a) learning a joint embedding model $M$ from which a $\mathcal{E}_{C_i}^i$ is extracted and utilized (b) during agent training to inform reward shaping and benefit agent learning, and (c) at deployment to provide end-users with $\mathcal{E}_{C_i}^i$ for agent actions.

## 4 State2Explanation (S2E) Framework

Figure 1 introduces our State2Explanation (S2E) framework which includes a three-step process to provide a benefit to both the RL agent and the end-user. Specifically, S2E learns a joint embedding model $M$ to align agent state-action pair $(s_i, a_i)$ to concept-based explanations $\mathcal{E}_{C_i}^i$ (Fig. 1a). S2E then leverages the learned $M$ to inform reward shaping during agent training (Fig. 1b). Finally, S2E leverages the learned $M$ to provide $\mathcal{E}_{C_i}^i$ about the trained agent's decision making to end-users at deployment (Fig. 1c). Below, we further detail S2E.

### 4.1 Joint Embedding Model $M$

A core component of S2E is learning a joint embedding model, $M$, to align an agent state-action pair, $(s_i, a_i)$ with an associated concept-based explanation $\mathcal{E}_{C_i}^i$. The motivation for learning $M$ is to explicitly map an agent's representation, $s_i$, into a representation understandable by our non-AI expert users, $\mathcal{E}_{C_i}^i$. Joint embedding models have been leveraged in many applications, such as in image captioning [55, 34], knowledge-graph generation [28, 43], and RL [42, 41, 50] to improve task learning. In the context of S2E, we demonstrate that joint embedding $M$ can provide the *dual* benefit to i) agent learning, and ii) improved end user understanding of agent actions. Our $M$ is inspired by [50] who learn an embedding space in the context of hierarchical RL tasks. However, unlike in [50], our $M$ is not constrained to hierarchical RL tasks, and we consider domains in which multiple states can be associated to multiple, non-unique concepts.

The input to our joint embedding model, $M$, is defined by $\langle [s_i, s_{i-1}], g_i, \mathcal{E}_{C_i}^i, y_i \rangle$, which includes an agent's current state $s_i$, previous state $s_{i-1}$, other contextual game information (i.e., player in a multiplayer game, whether game is over) $g_i$, associated concept-based explanation $\mathcal{E}_{C_i}^i$, and $y_i$ which defines if $s_i$ and $\mathcal{E}_{C_i}^i$ are aligned or misaligned. Note, including $s_i$ and $s_{i-1}$ implicitly encodes the

action $a_i$ taken to transition from $s_{i-1}$ to $s_i$. Our contrastive loss function (Eq. 1) is adapted from [50], in which $\theta$ represents the model's parameters, and $i \in N$ represents the $i^{th}$ training sample. Specifically, $M$ minimizes the L2-norm of the difference between the embedding vectors when $s_{embed}^i$ and $exp_{embed}^i$ are aligned ($y = 0$) and maximize the L2-norm of the difference when $s_{embed}^i$ and $exp_{embed}^i$ are misaligned ($y = 1$).

$$L(\theta) = \frac{1}{N} \sum_i^N (\|s_{embed}^i - \mathcal{E}_{embed}^i\| - y)^2 \tag{1}$$

To train $M$, we leverage a dataset $D = \{D^a, D^m\}$, in which $D^a$ and $D^m$ denote the set of aligned and misaligned data samples, respectively. To collect $D^a$, we leverage domain knowledge to annotate relevant $s_i$ with its aligned concept set $C_i$. The concept set $C_i$ is then produced into a natural language explanation, $\mathcal{E}_{C_i}^i$, manually via fixed, expert-defined templates. That is, each $C_i$ has one templated $\mathcal{E}_{C_i}^i$. The list of concepts for our evaluation domains are in Section 5.1. To collect $D^m$, each $s_i$ is paired with $z$ concepts *not* present in concept set $C_i$. Section 5.2 provides more details on $D$.

**Explanation Retrieval**   Given $M$, for any $(s_i, a_i)$, we extract the closest explanation embedding $exp_{embed}^i$ for corresponding $s_{embed}^i$, and decode $exp_{embed}^i$ to an $\mathcal{E}_{C_i}^i$ similar to image-to-text retrieval [31, 6]. Specifically, we rank the set of possible explanation embeddings, $\{exp_{embed}^i..exp_{embed}^k\}$, by their L2-norm distance to $s_i$. The decoding of the best ranked $exp_{embed}^i$ to $\mathcal{E}_{C_i}^i$ is then a vocabulary look-up. It is possible to utilize large language models (LLMs), and *generate* $\mathcal{E}_{C_i}^i$ from $exp_{embed}^i$ which we consider future work. Instead, in this work, we focus on validating the dual utility of embedding-based $\mathcal{E}_{C_i}^i$ retrieval to both the RL agent as well as end-user.

## 4.2   Reward Shaping in RL via Joint Embedding Model $M$

During agent learning, we utilize $M$ to retrieve the closest $\mathcal{E}_{C_i}^i$ associated with a $(s_i, a_i)$ for informing reward shaping. Prior work has shown that reward shaping can improve agent learning rate and sample efficiency [11, 25, 32]. However, designing an effective dense reward function is not trivial [16, 12], often requiring a two-stage trial-and-error procedure to 1) determine *when* a state-action pair deserves an intermediate reward, and 2) determine *what* reward value such state-action pair should be attributed. Many methods, such as inverse reinforcement learning, utilize learned, black box reward functions to minimize the trial-and-error process [37, 1, 57]. However, with black-box reward functions, we lose transparency of why a $(s_i, a_i)$ is rewarded. To balance the design-transparency trade-off of dense reward functions, we leverage $M$ to inform *when* a $(s_i, a_i)$ pair should be rewarded based on the retrieved $\mathcal{E}_{C_i}^i$, and as a result, remove the first step of the trial-and-error process. Specifically, if $M$ returns a meaningful $\mathcal{E}_{C_i}^i$ for a $(s_i, a_i)$ (e.g.,concept set $C_i \neq \emptyset$), then such $(s_i, a_i)$ represents concepts that influence the agent's goal, and should receive an intermediate reward.

Note, $M$ does not remove the trial-and-error process needed to determine *what* values each reward component should receive. Instead, we perform a hyperparameter sweep to assign shaping values to each concept component when an expert-determined, continuous shaping function is not derivable. Thus, the "Explanation-to-reward-shaping" module in S2E performs a look-up to assign a reward value based on the retrieved $\mathcal{E}_{C_i}^i$. In Section 5.3 we validate how leveraging $M$ to inform reward shaping can improve an agent's learning rate comparable to expert-defined shaping functions.

## 4.3   Concept-Based Explanations to End Users via Joint Embedding Model $M$

Once an RL agent is trained and deployed, we leverage $M$ to provide $\mathcal{E}_{C_i}^i$ about an agent's decision-making to non AI expert end-users. Recall from Section 3 that $\mathcal{E}_{C_i}^i$ can include multiple concepts (i.e. $|C_i| > 1$). We posit that in some cases, $\mathcal{E}_{C_i}^i$ with *too many* concepts may become ineffective in aiding end-user understanding. In fact, prior works discuss the challenge of presenting information at the "right" level of abstraction to end-users [46, 53]. To the best of our knowledge, there are no explicit guidelines for what information abstractions are beneficial in sequential decision making settings. From analyzing prior end-user friendly explanation techniques [9, 14], we infer two important dimensions to consider for explanation abstraction in sequential decision making: Information Filtering (InF), and Temporal Grouping (TeG).

**Information Filtering (InF)**   InF filters out concepts within $\mathcal{E}_{C_i}^i$ to only include those that immediately influence the goal. For example, consider a $\mathcal{E}_{C_i}^i$ in Lunar Lander, "Fire left engine because it brings lander closer to the center, decreases lander velocity to avoid crashing, and decreases tilt". Applying *InF* may produce a filtered explanation, "Fire left engine because it brings lander closer to the center and decreases tilt" given a $(s_i, a_i)$ when the lander needs to immediately correct its tilt and

| Connect 4 Concepts | Description |
|---|---|
| BW | Blocks an opponent win |
| CD | Provides center dominance |
| 3IR | Creates a three-in-a-row |
| 3IR_BL | Creates a three-in-a-row but blocked from a win |
| W | A four-in-a-row win |
| NULL ($C_i = \emptyset$) | Any neutral state that advances the game |

| Lunar Lander Concepts | Description |
|---|---|
| POS | Brings lander closer to center |
| VEL | Decrease lander speed to avoid crashing |
| TILT | Decrease tilt |
| RLEG | Encourage right leg contact with ground |
| LLEG | Encourage left leg contact with ground |
| MF | Conserve main fuel usage |
| SF | Conserve side fuel usage |
| L | Land without crashing |

$\varepsilon_{concept}$: Play column 4 because it provides center dominance and creates a three-in-a-row.

$\varepsilon_{concept}$: Fire left engine because it moves lander closer to the center, decreases lander speed to avoid crashing, decreases tilt of lander, and conserves main fuel usage.

Figure 2: The tables present concepts for Connect 4, derived from [3], and Lunar Lander, derived from [4], with accompanying example $\mathcal{E}^i_{C_i}$.

position to avoid a trajectory path leading to a crash. While the other concepts are important, they are filtered out since they do not effect the immediate chance of meeting $G$.

To perform InF, we leverage domain knowledge and expert-defined thresholding to determine which concepts in $C_i$ for $(s_i, a_i)$ critically contribute towards the agent's ability to reach goal $G$. Thresholds are determined via qualitative analysis of an RL agent's policy. In continuous domains, these threshold values are extracted by finding the values in the agent's state space that determine turning points in the agent's ability to reach goal $G$. For example, qualitative analysis may produce that an agent's $pos_x > 0.15$ is a critical turning point for the agent's ability to land close to the center, and if $pos_x > 0.15$, the agent's action is crucially linked with decreasing its $pos_x$. Thus, $C_{iw/InF}$ represents the concept set after thresholding is applied, in which concepts $c \in C_i$ that do not meet the determined thresholds are filtered out. As a result, an abstracted explanation, using InF, templates $C_{iw/InF}$ into a natural language explanation $\mathcal{E}^i_{C_{iw/InF}}$. In Appendix A, we provide more details on how concepts are filtered via qualitative analysis in InF and include an ablation study analyzing the sensitivity of chosen thresholds. Note, while we qualitatively determine thresholds, the InF method can be automated in future work.

**Temporal Grouping (TeG)**    TeG automatically groups explanations over a sequence of time that represents a consecutive pattern. For example, if in Lunar Lander, $\mathcal{E}^i_{C_i}$ and $\mathcal{E}^{i+2}_{C_i}$ is "Fire left engine because it decreases tilt" and $\mathcal{E}^{i+1}_{C_i}$ and $\mathcal{E}^{i+3}_{C_i}$ is "Fire right engine because it decreases tilt", applying TeG produces "For the next 4 steps, alternate firing left and right engine to decrease tilt".

To perform TeG, we analyze an agent's rollout at deployment to extract series of $\left(s_{[i,i+n]}, a_{[i,i+n]}\right)$ with a repeated pattern of concept sets $C_{[i,i+n]}$. Therefore, a grouped explanation provides a single $\mathcal{E}^i_{C_{iw/TeG}}$, that templates the repeated pattern within $C_{[i,i+n]}$ across the $n$ state-action pairs. We posit that TeG is likely to be important in domains where actions are sampled at high frequency (e.g. Lunar Lander or Robotics), requiring an abstraction over actions to provide meaningful explanations for consecutively alike state-action pairs.

Overall, the "Explanation Abstraction" component in S2E (see Fig. 1c) determines whether InF, TeG or both should be applied to $\mathcal{E}^i_{C_i}$ for providing an abstracted, concept-based explanation. In Section 6 we demonstrate the effectiveness of InF and TeG in high frequency domains, such as Lunar Lander, for improved end user understanding.

# 5 Model Evaluations

In this section, we perform quantitative evaluations to validate the joint embedding model $M$ in S2E by evaluating $M$'s explanation retrieval accuracy (Sec. 5.2), and $M$'s ability to inform reward shaping during agent training (Sec. 5.3).

## 5.1 Evaluation Domains

**Connect 4** is an adversarial board game in which the objective is to achieve a four-in-a-row, in any direction. The state space is a discrete, 2D array representation of token positions. Also, the action

space is discrete and each player selects the column to place a token in. **Lunar Lander** is a trajectory optimization problem in which the lander must land on a landing pad. We utilize LunarLander-v2 from OpenAI Gym [4]. The state space is a continuous 1D array including the lander's linear position and velocity, angular velocity, angle, and leg contact. The game has four discrete actions: fire left engine, fire right engine, do nothing, and fire main engine.

**Concept Collection**  Concepts for a domain may be extracted from expert players or commentators. However for Lunar Lander and Connect 4, there are no existing datasets for mining concepts. Thus, we leverage expert domain knowledge to attribute concepts to state-action pairs. Figure 2 lists the set of concepts we consider when attributing concepts to a $(s_i, a_i)$ in Connect 4 and Lunar Lander, as well as provides example $\mathcal{E}_{C_i}^i$. Concepts for Connect 4 are derived from the strategies outlined in [3]. Note, "NULL" describes all other $(s_i, a_i)$ that are not attributed to a concept. Alternatively, concepts for Lunar Lander are derived from the domain's existing reward function [4]. In this domain, the existing dense reward function includes the primary concepts important towards successful task completion. In Appendices B.1 and B.2, we provide an an expanded list of example $\mathcal{E}_{C_i}^i$, spanning all concepts derived for Connect 4 and Lunar Lander.

Note, for both domains, the concepts are defined via mathematical or logical representations of the state. For example, in Connect 4, the concept of "BW", blocking an opponent win, can explicitly be encoded by board representations where any three-in-a-row pattern is blocked with a token from the opponent player. Similarly, in Lunar Lander, the "POS", position concept, is modelled by mathematical representation of the agent's position over time. Deriving concepts via mathematical or logical representations allow us to automatically collect concepts from states, as well as use such mathematical or logical rules to evaluate that concepts and states are accurately paired. In many applications, it may be infeasible to derive mathematical or logical rules from a state representation. In these scenarios, concepts can be collected via crowd sourcing, [20], or obtained via "think-aloud" procedures [14].

## 5.2  Evaluation of Joint Embedding Models

Below we demonstrate high recall rates of the joint embedding models in S2E in both the Connect 4 and Lunar Lander domain.

**Datasets**  Recall from Section 4.1 that to train $M$ we utilize a dataset $D = \{D^a, D^m\}$. For Connect 4, $D_{C4}$ includes approximately 3 million samples, and for Lunar Lander, $D_{LL}$ includes approximately 2 million samples. We randomly sample $D_{C4}^a$ and $D_{LL}^a$ from an expert RL policy. We obtain $D_{C4}^m$ and $D_{LL}^m$, by selecting $z$ incorrect concepts from $C_{C4}$ and $C_{LL}$, as replacement for each sample in $D_{C4}^a$ and $D_{LL}^a$ (more detailed numbers in Appendix C).

**Model Architectures & Training**  $M_{C4}$ and $M_{LL}$ denote the joint embedding models for Connect 4 and Lunar Lander, respectively. The model architectures for $M_{C4}$ and $M_{LL}$ leverage LSTMs to learn explanation embedding $exp_{\text{embed}}^i$ for a given $\mathcal{E}_{C_i}^i$. For Connect 4, to learn $s_{\text{embed}}^i$ for $(s_i, a_i)$, we leverage Convolutional Neural Networks to learn local and spatial relationships between tokens on a 2D game board. For Lunar Lander, we leverage Fully Connected Networks to learn $s_{embed}^i$. More details (split, learning rate, etc) are in Appendix C.

**Results**  Similar to image-to-text retrieval [31, 6], we evaluate $M_{C4}$ and $M_{LL}$ via recall rate, at $k = \{1, 2, 3\}$, which evaluates whether the retrieved $\mathcal{E}_{C_i}^i$ is ranked in the top $k$. Figure 3a provides the average recall rates of $M_{C4}$ and $M_{LL}$, across 5 random seeds. We observe that $M_{LL}$ performs with near 100% accuracy, whereas $M_{C4}$ has an average recall rate of 88%. Given the lower recall rates by $M_{C4}$, in Figure 3b we examine the false positive and false negative explanation retrievals. We see that $M_{C4}$ has greatest challenge in correctly retrieving $\mathcal{E}_{C_i}^i$ with "BW" and "3IR_BL". We posit that "BW" and "3IR_BL" may occur in board states with greater variation in comparison to other concepts, leading to higher incorrect retrievals. Additional analyses of $M_{C4}$ and $M_{LL}$ are in Appendix C.4.

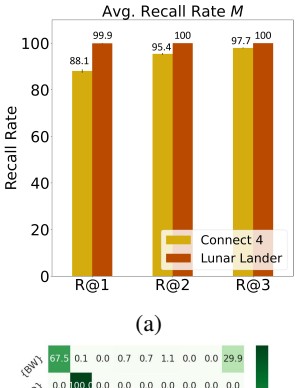

(a)

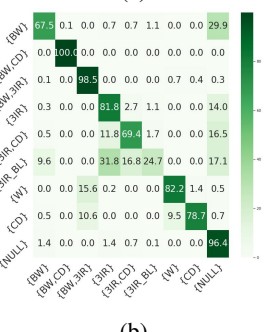

(b)

Figure 3: (a) recall@k of $M_{C4}$ and $M_{LL}$, and (b) confusion matrix for $M_{C4}$'s $\mathcal{E}_{C_i}$ retrievals.

Note, incorrect retrievals of $\mathcal{E}_{C_i}^i$ for a given $(s_i, a_i)$ *can* have a negative downstream impact within S2E. Specifically, when S2E is leveraged during the RL agent's training, incorrect retrievals of

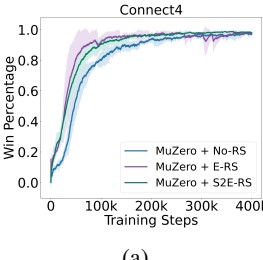 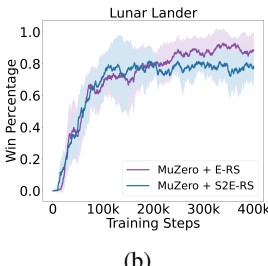 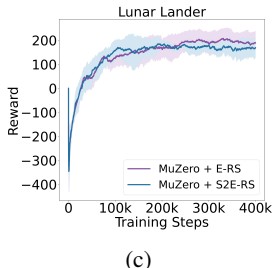

|  (a) | (b) | (c) |

Figure 4: $M_{C4}$ improves agent learning rate by ~200 training steps compared to SoTA agent without reward shaping (a), while $M_{LL}$ maintains similar agent learning rate to SoTA agent with expert-defined reward shaping (b and c).

$\mathcal{E}^i_{C_i}$ can incorrectly providing shaping rewards to the agent and in return impacting learned agent policy. Similarly, when S2E is leveraged at deployment to provide end-user understanding, incorrect retrievals of $\mathcal{E}^i_{C_i}$ can confuse end-users and hinder their understanding of the agent's behavior. However, in Section 5.3 our results demonstrate that the percentage of incorrect retrievals from the joint embedding models do not significantly impact the RL agent's learned policy. Similarly, in Section 6, we demonstrate that the percentage of incorrect explanation retrievals are not significantly detrimental to user performance. Nevertheless, it is important that the joint embedding models within S2E have high recall rate, especially when applied in high-stakes or mission critical scenarios.

## 5.3 Evaluation of $M$ for Reward Shaping in Agent Training

We validate that both joint embedding models in S2E, $M_{C4}$ and $M_{LL}$, inform reward shaping comparable to expert-defined dense, reward functions.

**RL Agent & Shaping Rewards**  Our S2E framework is model agnostic and $M$ does not make any assumptions on the type of RL model utilized. Given our domains are complex games, we leverage a state-of-the-art (SoTA) RL algorithm MuZero [47] to evaluate $M$'s ability to inform reward shaping. We use the open-source version of MuZero available in [52] (details in Appendix D.1).

Recall from Section 4 that $M_{C4}$ and $M_{LL}$ inform *when* to provide shaping rewards, but the shaping values are expert determined. For Lunar Lander, there exists a SoTA dense, reward function [4]. In Connect 4, to our knowledge, there is no SoTA dense, reward function. Thus, we perform a hyperparameter sweep to assign shaping values for each concept (values provided in Appendix D.2).

**Results**  To evaluate the efficacy of $M_{C4}$ and $M_{LL}$ in informing reward shaping, we measure the agents' *Reward* and *Win%*. In Lunar Lander, we evaluate $M_{LL}$'s ability to inform reward shaping **(MuZero+S2E-RS)** in comparison to a baseline MuZero agent that is informed by an existing expert-defined reward shaping **(MuZero+E-RS)**. Figure 4b and Figure 4c demonstrate that both the **MuZero+S2E-RS** and **MuZero+E-RS** agents achieve comparable *Reward* and *Win%*. While performance is comparable between both agents, in **MuZero+S2E-RS**, $M_{LL}$ provides an automated method to determine *when* a reward value should be presented, which otherwise has to be manually encoded, such as in **MuZero+E-RS** via an expert-defined function.

In Connect 4, since there does not exist a SoTA reward shaping function, we compare $M_{C4}$'s ability to inform reward shaping **(MuZero+S2E-RS)** against a baseline MuZero agent with sparse rewards **(MuZero+No-RS)**. Figure 4a shows that **MuZero+S2E-RS** has improved learning rate and requires ~200k less training samples to achieve similar *Win%* in comparison to **MuZero+No-RS**. We also evaluate **MuZero+S2E-RS** against an upper bound in which we provide expert-defined reward shaping **(MuZero+E-RS)** to analyze how incorrect retrievals from $M_{C4}$ may affect the agent's learning rate. Although $M_{C4}$ retrieves incorrect explanations 13.9% of times (see Appendix C.4), we observe that **MuZero+S2E-RS**'s learning rate is not significantly affected (Fig. 4a).

Overall, these results demonstrate the joint embedding models' ability to effectively inform reward shaping. In Connect 4, we demonstrate that, even with an imperfect joint embedding model, S2E can inform reward shaping and improve the agent's learning rate compared to the SoTA agent trained on sparse rewards. Similarly, in Lunar Lander, we demonstrate that our S2E informs reward shaping comparably to the existing SoTA agent that is trained with an expert-defined dense, reward function.

# 6 User Evaluation

In this section, we validate S2E with end-users, demonstrating that our retrieved $\mathcal{E}_{C_i}$ significantly improve user task performance in both Connect 4 and Lunar Lander.

**Study Procedure**  Participants performed an online study in which they played several games from the domains in four stages. Specifically, 1) *Practice* included playing 2 practice games, 2) *Pre-Test* included playing 3 scored games, 3) *Explanation* included interacting with an expert player (well-trained RL agent) with exposure to explanations from an assigned study condition, if applicable and 4) *Post-Test* included playing 3 more scored games. Details about the the study procedures are in Appendix E.1.

**Metrics**  We measure the difference between participant *Pre-Test* and *Post-Test Adjusted Task Score (ATS)* to analyze any task improvement with exposure to the study conditions. Participant *ATS* is defined by the expert-defined reward functions utilized by the RL agents during training (evaluated in Sec. 5.3, detailed in Appendix D.2). If $R(s, a)$ denotes the reward associated with a given $(s, a)$ and $N$ defines the total actions taken by the user in a game, then *ATS* is defined as a normalized aggregation of user rewards received in a game: $ATS = \sum_n^N R(s_n, a_n)/N$.

## 6.1 Connect 4 Study Specifics

**Study Conditions**  We conducted a five-way between-subjects study with the following conditions. Participants could receive: 1) **None (Baseline)**: no information about the agent's action to be played, 2) $\mathcal{E}_A$ **(Baseline)**: action-based explanation that contains the action to be played, 3) $\mathcal{E}_V$ **(Baseline)**: value-based explanation stating the action to be played along with the action's value in comparison to the values of all other actions for a given state, 4) **GT $\mathcal{E}_{C_i}$**: concept-based explanation using expert-defined, ground-truth concepts for a state-action pair, and 5) **S2E $\mathcal{E}_{C_i}$**: a concept-based explanation from S2E for a state-action pair. Note, given that Connect 4 has a discrete state space, and actions are sampled at low frequency, the domain does not need further abstracted $\mathcal{E}_{C_i}$ via InF and TeG. Additionally, $\mathcal{E}_V$ is similar to the action-cost condition in [48], as well as the Q-values presented per action in [27]. Note, precondition-based explanations in [48] require hierarchical domains, and causal-based explanations in [36] require a learned structured causal model which prevent direct comparison with our framework and domains.

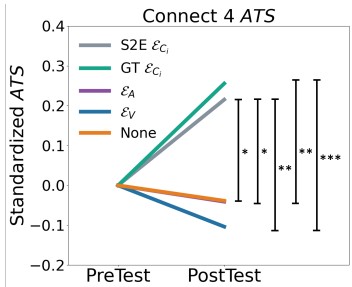

Figure 5: User adjusted task scores (ATS) in Connect 4. Statistical significance: * p < 0.05, ** p < 0.01, and *** p < 0.001; Details in Appendix E.3.

**Results**  The following results are from an IRB-approved study with participants recruited via Amazon Mechanical Turk (n=75, details in Appendix E.2). Our data is analyzed with a one-way ANOVA and a Tukey HSD post-hoc test, given that the assumptions of homoscedasticity (Levene's Test, p>0.3), and normality (Shapiro-Wilke, p>0.1) are met. Figure 5 shows a significant effect of explanation type on *ATS* (F(4, 70)=8.56, p<0.001). Specifically, we observe improvement in participant's *ATS* with exposure to our S2E $\mathcal{E}_{C_i}$, in comparison to *None* (t(70) = 3.08, p<0.05), $\mathcal{E}_A$ (t(70)=-3.25, p<0.05) and $\mathcal{E}_V$ (t(70)=4.03, p<0.01). Additionally, we observe similar *ATS* improvement when exposed to S2E $\mathcal{E}_{C_i}$ in comparison to GT $\mathcal{E}_{C_i}$. These results indicate the benefit of our S2E produced $\mathcal{E}_{C_i}$ in helping participants better understand Connect 4 and improve their *ATS*.

## 6.2 Lunar Lander Study Specifics

The underlying physics needed for Lunar Lander results in agent $(s_i, a_i)$ being sampled at high frequency and multiple, repetitive concepts being attributed to $(s_i, a_i)$ pairs. For example, "decrease velocity" is a valid concept for every $(s_i, a_i)$ until the end of a game. Recall from Section 4.3 that $\mathcal{E}_{C_i}$ may need to be further abstracted in these scenarios to provide meaningful explanations to end-users. Therefore, to evaluate the utility of abstracted concept-based explanations, we introduce 3 more study conditions that utilize our InF and TeG methods outlined in Section 4.3.

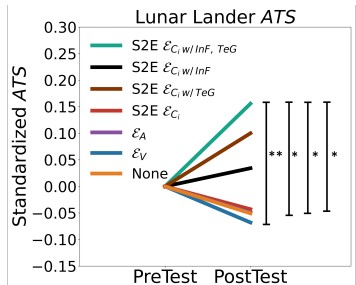

Figure 6: User ATS in Lunar Lander; * p < 0.05, ** p < 0.01

**Study Conditions**  We perform a seven-way between-subjects user study. Similar to Connect 4, we include **None**, $\mathcal{E}_A$, $\mathcal{E}_V$, and **S2E $\mathcal{E}_{C_i}$**. Unique to our Lunar Lander study participants

could receive 1) **S2E** $\mathcal{E}_{C_i \text{w/ TeG}}$: a concept-based explanation from S2E with additional abstractions performed using the TeG method, 2) **S2E** $\mathcal{E}_{C_i \text{w/ InF}}$: a concept-based explanation from S2E with additional abstractions performed using the InF method, and 3) **S2E** $\mathcal{E}_{C_i \text{w/ InF, TeG}}$: a concept-based explanation from S2E with additional abstractions performed using both TeG and InF.

**Results**   The following results are from an IRB-approved study with participants recruited via Amazon Mechanical Turk (n=105, details in Appendix E.2). Our data is analyzed with a one-way ANOVA and a Tukey HSD post-hoc test, given that the assumptions of homoscedasticity (Levene's Test, $p>0.7$), and normality (Shapiro-Wilke, $p>0.1$) are met. Figure 6 shows a significant effect of explanation type on participant *ATS* ($F(6,98)=3.67$; $p<0.01$). Specifically, we see significant improvement in participant's *ATS* with exposure to S2E $\mathcal{E}_{C_i \text{w/ InF, TeG}}$, in comparison to *None* ($t(98) = 3.15$, $p <0.05$)), $\mathcal{E}_A$ ($t(98)=-3.35$, $p<0.05$), $\mathcal{E}_V$ ($t(98)=3.59$, $p<0.01$) and $\mathcal{E}_{C_i}$ ($t(98)=-3.19$, $p<0.05$). These results demonstrate the need of our S2E abstraction methods when providing concept-based explanations to end-users, and that usage of both InF and TeG in high-frequency RL domains are crucial for *ATS* improvement.

# 7   Discussion and Conclusions

Our work introduces a unified framework, S2E, that involves learning a joint embedding model between agent state-action pairs and concept-based explanations to provide a dual benefit to the agent and end-user. We additionally outline a desiderata for what may constitute as a "concept" in sequential decision making problems, beyond the scope of prior concept-based explanations for sequential decision making. Our model evaluations demonstrate that the joint embedding model in S2E provides an automatic method for determining when reward shaping should be provided to improve agent learning rates. Our user evaluations demonstrate that concept-based explanations can significantly improve user task performance (Connect 4), but when considering high-frequency RL domains (Lunar Lander), the additional abstraction methods from S2E are important for producing abstracted concept-based explanations that significantly improve user task performance.

**Limitations:**   We present several areas of future work that aims to improve the generalizability of S2E. For instance, the concept-based explanations in our work are derived from mathematical representations and expert knowledge in each domain (see Sec. 5.1). To expand the generalizability of the S2E method to other complex domains, such as Robotics or open-world games, future work should explore how to collect and extract concepts in scenarios where mathematical representations of concepts may be infeasible. A future direction may include collecting expert commentary [20] or think-aloud [14] sessions from which concepts are automatically derived. Additionally, in many real-world applications, having access to large amounts of data for a domain may be infeasible. Thus, future work should also explore how to adapt the joint embedding model within S2E for few shot learning. Additionally, for application to high-stakes domain, future work should explore how to remediate incorrect explanation retrievals from the joint embedding model within S2E. While the small percentage of incorrect explanation retrievals do not significantly impact user task performance and agent learning in our tested domains, inaccurate explanation retrievals may have significant effects in mission-critical tasks. It is also important to highlight that our current concept-based explanations follow a fixed template for each unique concept set (see Appendices B.1, B.2). However, there are multiple ways of explaining the same concept, and future work includes learning a joint embedding model with greater language variety in the explanations combined with automatically generating such templates using language models. Furthermore, the InF method within S2E utilizes manually defined thresholds to produce abstracted concept-based explanations, and future work entails developing an automated InF method.

**Broader Impacts:**   S2E shows promise in applying the Protégé Effect to Human-AI Interaction, and that explanations of agent behavior are beneficial to both agents and users. These insights may promote AI agent architects to consider the utility of self-explaining agents in accelerating learning as well as providing transparency. Additionally, our work shows promise in using concept-based explanations of well-trained AI agents as a teaching tool, helping users improve their task performance. A potential risk of S2E is purposely training a joint embedding model to associate misleading concept-based explanations with state-action pairs. This could lead to a badly performing agent, and the agent providing deceitful explanations of its actions. Future work in explaining black-box AI systems should explore how to detect and prevent deceptive explanations.

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

# A Information Filtering (InF) Details

Recall from Section 4.3 that we utilize expert-defined thresholding and domain knowledge to determine which concepts within $C_i$ for a $(s_i, a_i)$ critically contribute towards the agent's ability to reach goal $G$. Specifically, these thresholds are expert-defined upper and lower bounds on the agent's state values that denote the agent's ability to succeed or fail in its goal. In our InF method, these thresholds are not mathematically derived, but are derived from RL-expert analysis. For a given domain, an RL expert visualizes multiple policy rollouts and analyze the different state values over time to manually determine the upper and lower bounds (turning points) that influence the agent's ability to reach G. Below we demonstrate how thresholds in InF for one of our evaluation domains, Lunar Lander.

The concepts relevant to Lunar Lander include, position, velocity, tilt, side and main fuel, right and left leg contacts, and landing, and these concepts are derived from the existing, expert-defined, dense reward function [4] (see Section 5.1). Within the dense, reward function, binary concept components are rewarded via constant shaping values, while the continuous concepts are rewarded through a continuous shaping function (see Appendix D.2). In our work, we consider applying thresholding to concepts defined via a continuous function since these scenarios typically result in multiple concepts being rewarded over multiple time steps, and are more likely to warrant information filtering.

In the context of Lunar Lander, Figure 7 and Figure 8 show the two concepts on which thresholding is applied when examining a well trained agent's policy, specifically the lander's x-position and tilt. Recall that thresholds for InF can be determined by examining an agent's state values to identify turning points in the agent's ability to reach $G$. In Figure 7, we extract that a suitable threshold for the x-position concept is 0.15. We identify that such x-position value corresponds to the time step at which the agent starts firing its left engine to critically correct its increasing position value. Similarly, in Figure 8, we extract that suitable thresholds for the tilt concept are 0.01 and -0.05. We find that such tilt values correspond with frequently alternating firing of left and right engine to critically correct an increasing tilt value. We do not perform a thresholding analysis on velocity, as with domain analysis we observe that decreasing the agent's velocity is directly associated with firing the main engine and is performed to prevent crashing. Additionally, we do not consider the agent's y-position for thresholding as the agent follows a steady, downward linear trajectory, and visible turning points cannot be extracted by qualitative analysis. While we qualitatively determine thresholds, the InF method can be automated in future work.

Overall, in our InF method, if $C_i$ denotes the original concept set for a given $(s_i, a_i)$, then $C_{i\text{w/InF}}$ represents the concept set after thresholding is applied, in which concepts $c \in C_i$ that do not meet the threshold are filtered out. As a result, an abstracted explanation, using InF, templates $Ct_i$ into a natural language explanation $\mathcal{E}^i_{C_{i\text{w/InF}}}$.

## A.1 Thresholding Sensitivity Analysis

As stated above, the thresholds are manually derived from RL-expert analysis. Specifically, an RL expert visualizes multiple policy rollouts and analyzes the different state values over time to determine approximate upper and lower bounds (turning points) that influence the agent's ability to reach $G$. Given the thresholding approach in InF, different threshold values can result in filtering of different concepts and therefore produce different abstracted, concept-based explanations. To study the sensitivity of our chosen thresholds in Lunar Lander, we analyze how different threshold values for each concept affect the number of concepts filtered in a policy rollout.

Figure 9 demonstrate what fraction of concepts are filtered (y-axis) as the threshold values change (x-axis). When looking at Figure 9a, analyzing threshold values for the x-position concept, we see our chosen value is within the elbow of the curve, denoting that the rate of filtration of the concept slows down after 0.15. Similarly, when analyzing the threshold values for the tilt concept in Figure 9b and Figure 9c, we see the lower and upper bound of the tilt thresholds are also within the "elbow" of each curve. Note, when analyzing the upper bound threshold for the tilt concept (Figure 9b), the lower bound tilt threshold is fixed. Similarly, when analyzing the lower bound threshold for the tilt concept (Figure 9c), the upper bound tilt threshold is fixed. Overall, our threshold values being within the elbow of each curve provide some validation on their soundness. However, to study the complete impact of varying thresholding values for each concept, additional user studies with abstracted concept-based explanations produced from varying threshold values are required. We

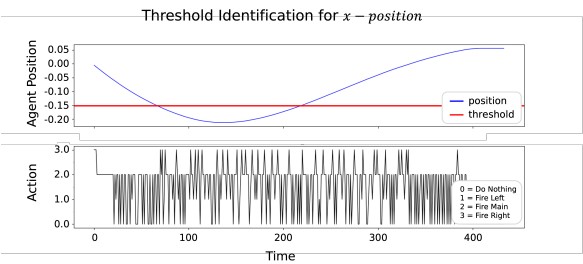

Figure 7: The position (top) and corresponding actions (bottom) of a trajectory sampled from a well-trained Lunar Lander agent policy.

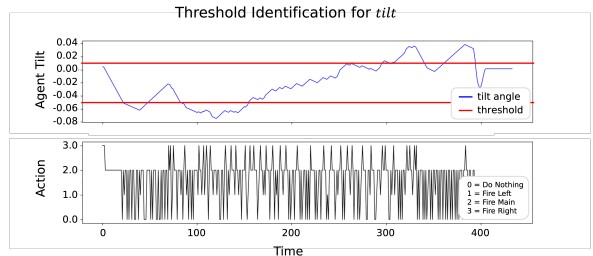

Figure 8: The tilt angle (top) and corresponding actions (bottom) of a trajectory sampled from a well-trained Lunar Lander agent policy.

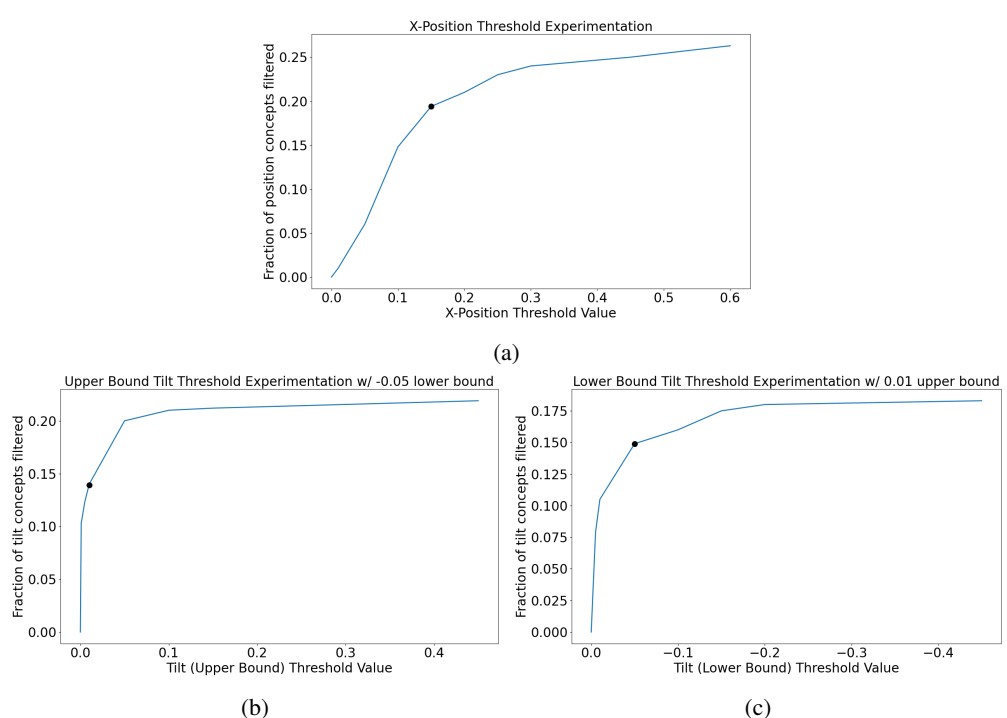

Figure 9: Sensitivity analysis on how the rate of concept filtration changes across various threshold values for Lunar Lander concepts of x-position (a) and tilt (b and c).

consider such analyses beyond the scope of our work, and one to further explore when considering future methods on automating the information filtering submodule within our S2E framework.

# B  Concept-Based Explanations for Experimental Domains

Below we present an exhaustive list of the different $\mathcal{E}^i_{C_i}$, utilized for each domain, paired with an instance of a $(s_i, a_i)$ pair for context. Note, the rows denote the possible concept sets $C_i$, and their corresponding fixed, templated explanations starting with "because". The action $a_i$ is always appended to the beginning of the explanation.

## B.1  Connect 4

| $(s_i, a_i)$ | $C_i$ | Corresponding $\mathcal{E}^i_{C_i}$ |
|---|---|---|
|  | {3IR} | "Play column 4 because it creates a three-in-a-row." |
|  | {3IR_BL} | "Play column 7 as a neutral move, it creates a three-in-a-row that is blocked by the opponent from a win." |
|  | {3IR, CD} | "Play column 4 because it provides center dominance and creates a three-in-a-row." |
|  | {BW} | "Play column 6 because it blocks the opponent from a win" |
|  | {BW, 3IR} | "Play column 5 because it creates a three-in-a-row and blocks the opponent from a win." |
|  | {BW, CD} | "Play column 4 because it provides center dominance and blocks the opponent from a win." |
|  | {CD} | "Play column 4 because it provides center dominance" |
|  | {W} | "Play column 7 because it leads to a four-in-a-row win" |
|  | {NULL} | "Play column 6 as a generic move not tied to a particular strategy" |

Table 1: Example state action pairs with associated concept lists and concept-based explanations for the Connect 4 domain.

## B.2 Lunar Lander

| $(s_i, a_i)$ | $C_i$ | Corresponding $\mathcal{E}_{C_i}^i$ |
|---|---|---|
|  | {POS, VEL, TILT, SF} | "Fire main engine because it moves lander closer to the center, decreases lander speed to avoid crashing, decreases tilt of lander, and conserves side fuel usage." |
|  | {POS, VEL, TILT, MF} | "Fire side engine because it moves lander closer to the center, decreases lander speed to avoid crashing, decreases tilt of lander, and conserves main fuel usage." |
|  | {POS, VEL, TILT} | "Do nothing because it moves lander closer to the center, decreases lander speed to avoid crashing, and decreases tilt of lander." |
|  | {POS, VEL, TILT, LLEG, SF} | "Fire main engine because it moves lander closer to the center, decreases lander speed to avoid crashing, decreases tilt of lander, encourages left leg contact, and conserves side fuel usage." |
|  | {POS, VEL, TILT, LLEG, MF} | "Fire side engine because it moves lander closer to the center, decreases lander speed to avoid crashing, decreases tilt of lander, encourages left leg contact, and conserves main fuel usage." |
|  | {POS, VEL, TILT, LLEG} | "Do nothing because it moves lander closer to the center, decreases lander speed to avoid crashing, decreases tilt of lander and encourages left leg contact." |
|  | {POS, VEL, TILT, RLEG, SF} | "Fire main engine because it moves lander closer to the center, decreases lander speed to avoid crashing, decreases tilt of lander and encourages right leg contact, and conserves side fuel." |
|  | {POS, VEL, TILT, RLEG, MF} | "Fire side engine because it moves lander closer to the center, decreases lander speed to avoid crashing, decreases tilt of lander, encourages right leg contact, and conserves main fuel." |
|  | {POS, VEL, TILT, RLEG} | "Do nothing because it moves lander closer to the center, decreases lander speed to avoid crashing, decreases tilt of lander, and encourages right leg contact." |
|  | {POS, VEL, TILT, LLEG, RLEG, SF} | "Fire main engine because it moves lander closer to the center, decreases lander speed to avoid crashing, decreases the tilt of the lander, encourages left and right leg contact and conserves side fuel." |
|  | {POS, VEL, TILT, LLEG, RLEG, MF} | "Fire side engine because it moves lander closer to the center, decreases the lander speed to avoid crashing, decreases the tilt of the lander, encourages left and right leg contact, and conserves main fuel." |

| | {POS, VEL, TILT, LLEG, RLEG} | "Do nothing because it moves lander closer to the center, decreases the lander speed to avoid crashing, decreases the tilt of the lander, encourages left and right leg contact." |
|---|---|---|
| | {L} | "Do nothing because it results in a land." |

Table 2: Example state action pairs with associated concept lists and concept-based explanations for the Connect 4 domain.

# C   Joint Embedding Model Details

## C.1   Dataset Details

**MisAligned and Aligned Data Breakdown**   Recall from Section 5 that $D_{C4}$ represents the dataset used for training and evaluating $M$ in Connect 4. Specifically, $D_{C4}$ includes $2,948,742$ total samples, in which $D_{C4}^a = 327,638$ aligned samples and $D_{C4}^m = 2,621,104$ misaligned samples. Similarly, $D_{LL}$ represents the dataset used for training and evaluating $M$ in Lunar Lander. Specifically, $D_{LL}$ includes $1,941,042$ total samples, in which $D_{LL}^a = 323,507$ aligned samples and $D_{LL}^m = 1,617,535$ misaligned samples. Recall that to create the misaligned dataset, $D^m$, we utilize $z$ incorrect concept sets available in each domain as replacement for the existing correct concepts utilized in $\mathcal{E}_{C_i}^i$ for each $d_i \in D^a$. Specifically, for Connect 4, $z = 8$ and we leverage all possible non-aligned concept sets for perturbation. In Lunar Lander, $z = 5$ and we randomly choose five non-aligned concept sets for perturbation. We randomly choose 5 of 13 possible concept sets to generate misaligned sample pairs per state to avoid a larger than necessary dataset. As a refresher, the list of possible concept sets for each domain are presented in Figure 2.

**Concept Breakdown in Aligned Data**   Table 3 and Table 4 provide a breakdown of the occurrences of each concept set within the training data for the joint embedding models, $M_{C4}$ (Connect 4) and $M_{LL}$ (Lunar Lander). We specifically show the breakdown in the aligned datasets, $D_{C4}^a$ and $D_{LL}^a$, given that such data is the raw data collected via rollout simulations in each domain. Recall that the additional misaligned data in the training set, $D_{C4}^m$ and $D_{LL}^m$, are synthetically generated via perturbations, and used for the contrastive learning. When the misaligned data is included in the training set, given the perturbations, the number of occurrences of each concept (aligned or misaligned) become equal.

| Concept Set | # of Samples in $D_{C4}^{aligned}$ |
|---|---|
| {3IR} | 43,896 |
| {3IR_BL} | 21,911 |
| {3IR, CD} | 6,901 |
| {BW} | 25,352 |
| {BW, 3IR} | 10,668 |
| {BW, CD} | 4,940 |
| {CD} | 30,591 |
| {W} | 23,137 |
| {NULL} | 160,242 |

Table 3: Concept set occurrence analysis of the aligned data within the training set used to train $M_{C4}$.

## C.2   Model Architectures

Figure 10 shows the model architecture for $M_{C4}$. Recall that the inputs to $M_{C4}$ includes $\langle [s_i, s_{i-1}], g_i, \mathcal{E}_{C_i}^i, y_i \rangle$. In the diagram, we don't explicitly show $y_i$ but it is used to denote whether

| Concept Set | # of Samples in $D_{LL}^{aligned}$ |
|---|---|
| {POS, VEL, TILT, MF} | 136,917 |
| {POS, VEL, TILT, SF} | 62,140 |
| {POS, VEL, TILT} | 53,944 |
| {POS, VEL, TILT, LLEG, MF} | 1,610 |
| {POS, VEL, TILT, LLEG, SF} | 2,981 |
| {POS, VEL, TILT, LLEG} | 4,219 |
| {POS, VEL, TILT, RLEG, MF} | 1,523 |
| {POS, VEL, TILT, RLEG, SF} | 2,399 |
| {POS, VEL, TILT, RLEG} | 3,228 |
| {POS, VEL, TILT, LLEG, RLEG, MF} | 1,730 |
| {POS, VEL, TILT, LLEG, RLEG, SF} | 12,026 |
| {POS, VEL, TILT, LLEG, RLEG} | 39,981 |
| {L} | 809 |

Table 4: Concept set occurrence analysis of the aligned data within the training set used to train $M_{LL}$.

$s_i$ and $\mathcal{E}_{C_i}^i$ are aligned or misaligned (see Section 4). The game boards ($s_i$ and $s_{i-1}$) in Connect 4 are represented as 6x7 2D array which are passed into two consecutive CNNs after padding to 7x7 2D arrays. The first set of CNNs (CNN1 and CNN3) for $s_i$ and $s_{i-1}$ have the following parameters: input_channels=1, output_channels=4, kernel_size= (3,3), stride=1. The second set of CNNs (CNN2 and CNN4) have the following parameters: input_channels=4, output_channels=6, kernel_size= (3,3), stride=1. The embedding outputs of CNN2 and CNN4 are then concatenated with the game information $g_i$, which includes boolean representations of the player/opponent as well as whether the game is over. The concatenated embedding is then passed through three FCNs. Specifically, FCN1's output dimension is 64. FCN2's output dimension is 32. FCN3's output dimension is 16. The output of FCN3 represents our final state embedding, $s_{embed}^i$. The concept-based explanation $\mathcal{E}_{C_i}^i$ is first passed through an embedding layer and then passed to a LSTM network to produce our $exp_{embed}^i$. The input size to the LSTM is of dimension 32, and output dimension is 16.

Figure 11 shows the model architecture for $M_{LL}$. The inputs to $M_{LL}$ includes $\langle [s_i, s_{i-1}], g_i, \mathcal{E}_{C_i}^i, y_i \rangle$. In the diagram, we don't explicitly show $y_i$, but it is used to denote whether $s_i$ and $\mathcal{E}_{C_i}^i$ are aligned or misaligned (see Section 4). The game boards ($s_i$ and $s_{i-1}$) in Lunar Lander are represented as 1x10 arrays which are passed into two consecutive FCNs. Specifically, the game board representation includes the 1x8 game state representation from OpenAI Gym [4], along with an inclusion of whether the side fuel or main fuel is in use. The first set of FCNs (FCN1 and FCN3) for $s_i$ and $s_{i-1}$ have an output dimension of 64. The second set of FCNs (FCN2 and FCN4) have an output dimension of 32. The embedding outputs of FCN2 and FCN4 are then concatenated with the game information $g_i$, which includes a boolean representation of whether the game is over. The concatenated embedding is then passed through three more FCNs Specifically, FCN5 has an output dimension of 16. FCN6 has an output dimension of 8. Similar to the architecture for $M_{C4}$, the concept-based explanation $\mathcal{E}_{C_i}^i$ is first passed through an embedding layer and then passed to a LSTM network to produce our $exp_{embed}^i$. The input size to the LSTM is of dimension 32, and output dimension is 8.

### C.3 Training Details

To train and evaluate $M_{C4}$ and $M_{LL}$, we utilize a 60%-20%-20% train-valid-test split on $D_{C4}$ and $D_{LL}$ (see total dataset size in Appendix C.1). The models are trained with learning rate of 0.001, batch size of 128, Adam Optimizer, and trained with 10 epochs. To train $M_{C4}$ and $M_{LL}$, we utilize a desktop computer with a NVIDIA GTX 1060 6GB GPU and an Intel i7 processor. The runtime for training and testing $M_{C4}$ and $M_{LL}$ takes approximately 30 minutes. See link to code in Appendix F.

### C.4 Additional Evaluations

**Learned Embedding Spaces of $M_{C4}$ and $M_{LL}$** In Figure 12a and Figure 12b, we provide a visualization of our learned joint embedding models, $M_{C4}$ and $M_{LL}$. The TSN-E visualizations is created using the models' training dataset. We do not draw any conclusions via these visualizations, but present them as an intuitive way to interpret our high dimensional, learned embedding spaces.

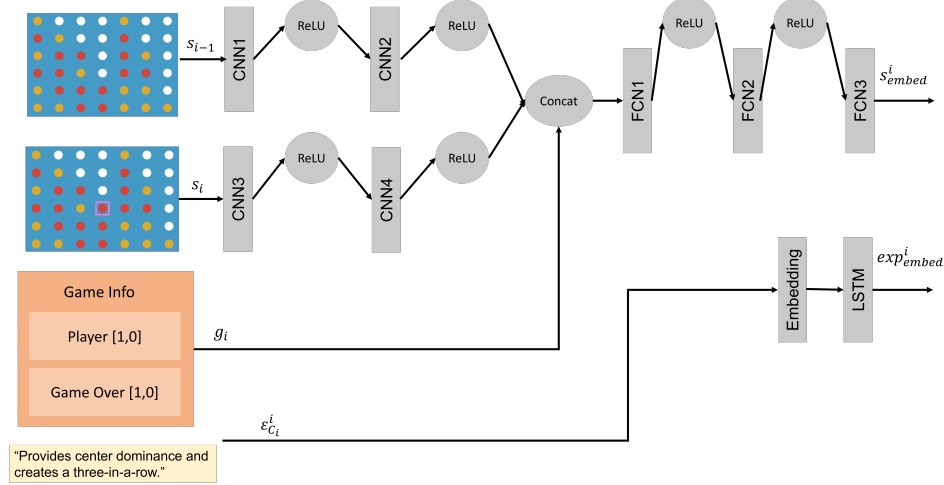

Figure 10: Joint Embedding Model $M$ Architecture for Connect 4.

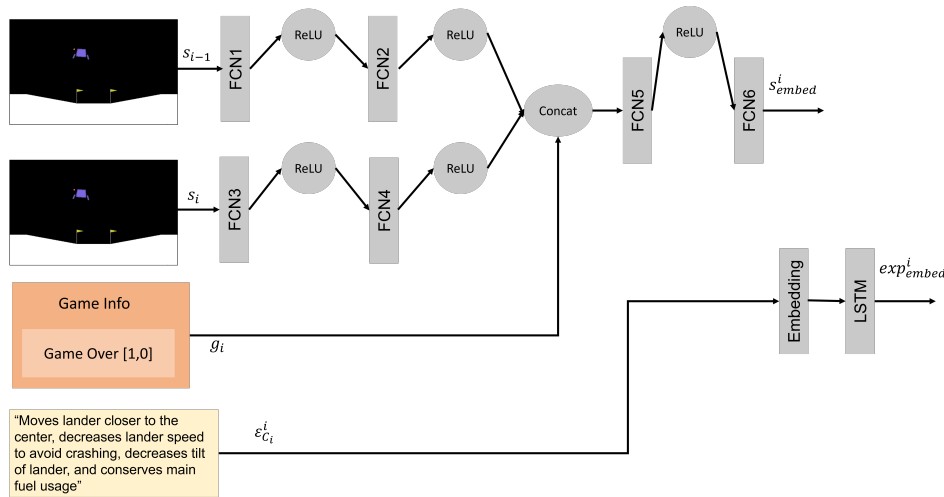

Figure 11: Joint Embedding Model $M$ Architecture for Lunar Lander.

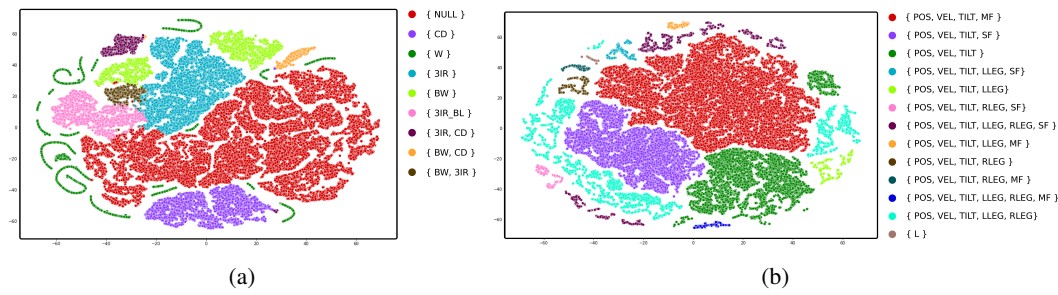

Figure 12: (a) and (b) demonstrate visualizations of the learned embedding spaces of the joint embedding models learned for Connect 4,$M_{C4}$, and Lunar Lander, $M_{LL}$.

**In the "Wild" Evaluation of** $M_{C4}$ **and** $M_{LL}$    We performed an evaluation of how well $M_{C4}$ and $M_{LL}$ continue to perform when utilized during an RL agent's training to inform reward shaping. Specifically, we analyzed the model's Recall@1, examining how often $\mathcal{E}^i_{C_i}$ was incorrectly retrieved for a given $(s_i, a_i)$. For Lunar Lander, $M_{LL}$'s average Recall@1, across the 5 seeds, during the

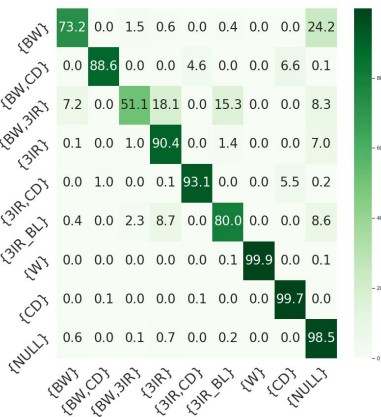

Figure 13: Evaluation of $M_{C4}$'s ability to retrieve correct $\mathcal{E}_{C_i}^i$ during the RL agent's training.

agent's training continued to be 100%. Figure 13 provides an analysis of the Connect 4 joint embedding model's, $M_{C4}$, average Recall@1. We see on average, $M_{C4}$ has an average Recall@1 of 86.1%, with 13.9% incorrect retrievals. As mentioned in Section 5.3 results, these incorrect retrievals do not significantly affect $M_{C4}$'s ability to inform reward shaping and the agent's learning rate.

## D  RL Agent Details

### D.1  RL Model Training

We utilized the open-source version of MuZero from [52], and utilize the provided hyperparameters. Please see Muzero's Github Repo from [52] [1], as well as our code linked in Appendix F, for more details. To train and validate our MuZero agents we leverage an NVIDIA GTX 1060 6GB GPU and an Intel i7 processor. The runtime for training a Connect 4 MuZero agent to around 400k training steps takes approximately 30 hours. The runtime for training a Lunar Lander MuZero agent to around 400k training steps takes approximately 9 hours.

### D.2  Shaping Rewards Per Concepts

For Lunar Lander, we utilize the existing dense reward function to determine the relevant concepts as well as their shaping values. Therefore each $c \in C_{LL}$ has an expert-defined shaping value. The existing shaping values can be found in the *LunarLander.py* provided by OpenAI Gym [4]. Table 5 summarizes the continuous functions outlined in [4] that we use to reward each concept $c \in C_{LL}$. Alternatively, there are no existing SoTA reward shaping values for the game of Connect 4. Therefore we perform a hyperparameter sweep to determine an optimal set of shaping values, and Table 6 summarizes our utilized shaping values for each concept $c \in C_{C4}$.

| Lunar Lander Concept | Shaping Values |
|---|---|
| POS (position) | $-100(\sqrt{(pos_x)^2 + (pos_y)^2})$ |
| VEL (velocity) | $-100(\sqrt{(vel_x)^2 + (vel_y)^2})$ |
| TILT (tilt) | $-100(\lvert angle_{lander} \rvert)$ |
| RLEG (right leg contact) | $10(bool_{RLEG})$ |
| LLEG (left leg contact) | $10(bool_{LLEG})$ |
| MF (main fuel) | $-0.3(bool_{main})$ |
| SF (side fuel) | $-0.03(bool_{side})$ |
| L (land) | $100$ |

Table 5: Corresponding shaping values from [4] for each Lunar Lander concept.

---

[1]https://github.com/werner-duvaud/muzero-general

| Connect 4 Concept | Shaping Values |
|---|---|
| 3IR (Three-in-Row) | 1 |
| 3IR_BL (Three-in-Row Blocked from Win) | 0 |
| CD (Center Dominance) | 1 |
| BW (Block Win) | 5 |
| W (Win) | 10 |
| NULL (Neutral State) | 0 |

Table 6: Corresponding shaping values determined from hyperparameter sweep for each Connect 4 concept.

# E   User Study Details

## E.1   Procedure Details

Participants performed an online study in which they were asked to play several games from the domain. The user studies for each domain were was broken into the following four stages: *Practice*, *Pre-Test*, *Explanation*, *Post-Test*. First, prior to the *Practice* stage, participants were introduced to the game including background information, game rules, and their task. Then in the *Practice*, participants played 2 practice games to get further acquainted with the specified domain. In *Pre-Test*, participants played 3 more games, this time being aware that their games were being scored. In *Explanation*, participants interacted with an expert player (well-trained RL agent) and were exposed to explanations about the player's actions via their assigned study condition. Note, participants in the "None" study condition did not receive any accompanying explanations during the *Explanation* stage. Specific to Connect 4, this stage consisted of observing the expert player (RL agent) play 3 games. In Lunar Lander, the *Explanation* stage consisted of observing the expert player (RL agent) play 1 game. In the *Post-Test*, participants played 3 more scored games, similar to the *Pre-Test*. After the *Post-Test* stage, participants completed a post-questionnaire survey that collected demographics data as well as their perceived task performance and their perceived utility of their explanation condition.

Some specifics unique to each domain include that for Connect 4, participants always played first, and their opponent player was a well-trained MuZero agent. Additionally, for Lunar Lander, to reduce the impact of learning effects, the starting position and initial force applied to the lander was randomized across all 8 games; however, the random combinations remained fixed across participants. Also, participants played the version of Lunar Lander implemented by OpenAI Gym [4].

## E.2   Participant Information

**Connect 4**   We recruited 84 participants from Amazon Mechanical Turk, for an IRB-approved study and participants were required to be novice players of Connect 4. Of the 84 participants, 9 were filtered out for not finishing the study. The remaining 75 participants, 15 per study condition, included 40 males and 35 females, all over the age of 18 (M=31.57, SD=7.12). The study took on average 20 minutes, and participants were compensated $5.00.

**Lunar Lander**   We recruited 118 participants from Amazon Mechanical Turk, for an IRB-approved study. Participants were required to be novice players of Lunar Lander. Of the 118 participants, 13 were filtered out for not finishing the study or demonstrating no effort. The remaining 105 participants, 15 per study condition, included 50 males and 55 females, all over the age of 18 (M=30.57, SD=6.75). The study took on average 20 minutes, and participants were compensated $5.00.

## E.3   Additional Evaluation Details

**Participant ATS**   Note, the adjusted task score *(ATS)* adjusts participant's *Post-Test* average task performance by their *Pre-Test* average task performance. Note, this adjustment does not change any of the results or statistical analyses and is purely for visualization purposes. Specifically, we perform the adjustment to visualize the relative changes among participant ATS using a common starting point.

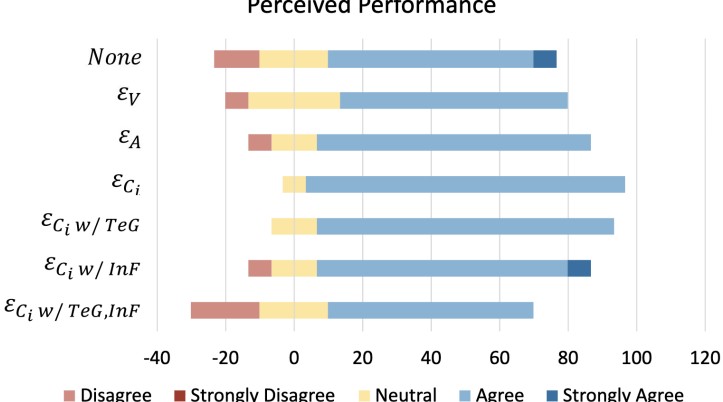

Figure 14: Analysis of perceived performance *Perf* shows participants across all conditions believed their performances improved after exposure to the expert player and assigned explanations.

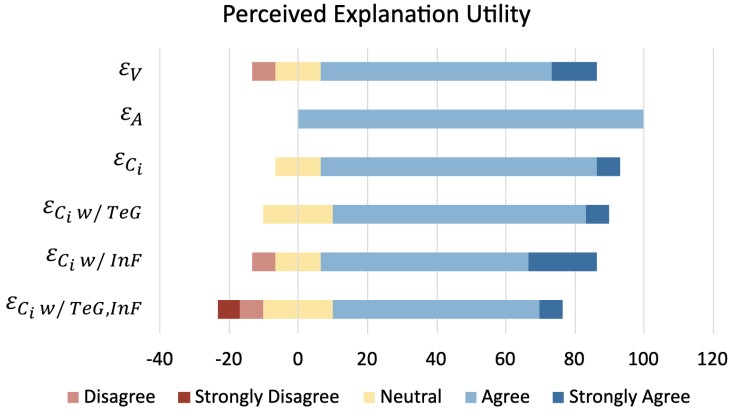

Figure 15: Analysis of perceived explanation utility *EU* shows participants across all explanation conditions believed their assigned study condition improved task understanding.

Additionally, to keep the *ATS* graphs legible, we present the mean and standard deviations for each study condition here, as opposed to including error bar lines on the graphs. The following are the mean and standard deviations in the Connect 4 user study conditions. Specifically, (1) *None* (**Baseline**): M=-0.039, SD=0.222, (2) $\mathcal{E}_A$ (**Baseline**): M=-0.041, SD=0.114 (3) $\mathcal{E}_V$ (**Baseline**): M=-0.104, SD= 0.346 (4) **GT** $\mathcal{E}_{C_i}$: M=0.256, SD=0.318 (5) **S2E** $\mathcal{E}_{C_i}$: M=0.215, SD=0.194. The following are the mean and standard deviations in the Lunar Lander user study conditions. Specifically (1) *None* (**Baseline**): M=-0.051, SD=0.191, (2) $\mathcal{E}_A$ (**Baseline**): M=-0.049, SD=0.136 (3) $\mathcal{E}_V$ (**Baseline**): M=-0.067, SD=0.211 (4) **S2E** $\mathcal{E}_{C_i}$: M=-0.043, SD=0.192, (5) **S2E** $\mathcal{E}_{C_i \text{w/ TeG}}$: M=0.100, SD=0.154 (6) **S2E** $\mathcal{E}_{C_i \text{w/ InF}}$: M=0.034, SD=0.190, (7) **S2E** $\mathcal{E}_{C_i \text{w/ InF, TeG}}$: M=0.156, SD=0.182.

**User Perception Analyses**  Below we present additional evaluations performed on user responses in our questionnaire. Specifically, we present participant's perceived performance *Perf* and participant's perceived explanation utility *EU*. Note, *Perf* is measured on a 5-point Likert scale to the following question "I believe my game performance improved after witnessing the expert play and seeing the expert's provided reasoning". The *EU* metric is measured on a 5-point Likert scale to the following question "I believe the expert's provided reasoning helped improve my understanding of the game", and is only measured for the explanation conditions. From both Figure 14 and Figure 15 we observe that participants across all study conditions had high agrees for *Perf* and *EU*, despite their being significant differences in their objective task performance. These qualitative results demonstrate the Dunning-Krunger Effect [13], in that users with low expertise in an area often overestimate their own performance or knowledge.

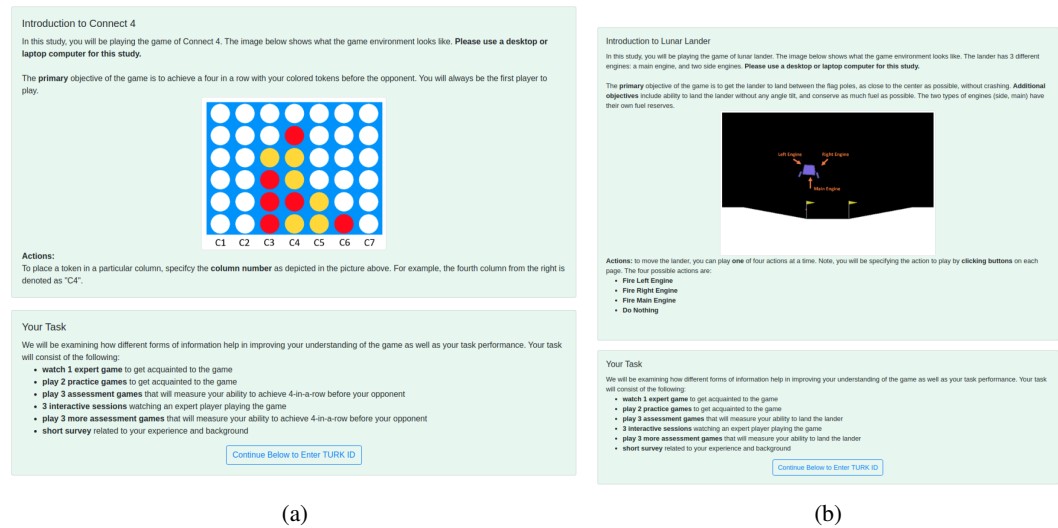

(a)                  (b)

Figure 16: Participants saw the following as introductions to the task for (a) Connect 4 and (b) Lunar Lander.

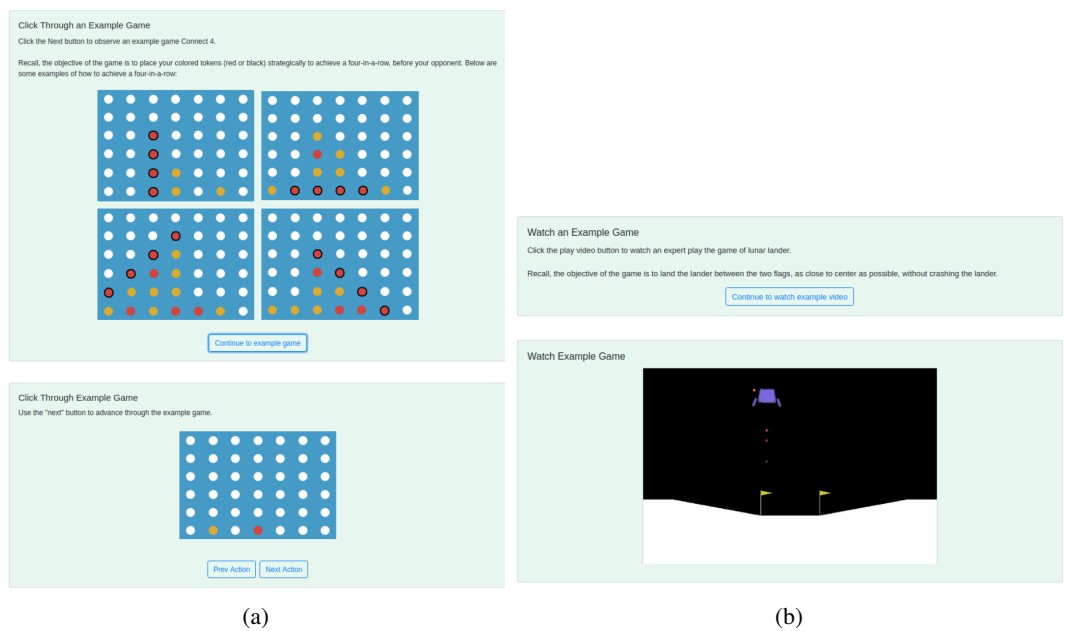

(a)                  (b)

Figure 17: Participants saw the following as introductions to the game for (a) Connect 4 and (b) Lunar Lander.

## E.4 Visuals of User Study

In this section, we provide visuals of the user interface for each user study.

**Introduction Stage** In both user studies, after receiving consent of participation, all participants were given an introduction to both the user study as well as the specific domain of the study. Figure 16a(a)and Figure 16b(b) show the introductions participants received for Connect 4, while Figure 17a(a) and Figure 17b(b) show the introductions participants received for Lunar Lander. Note the

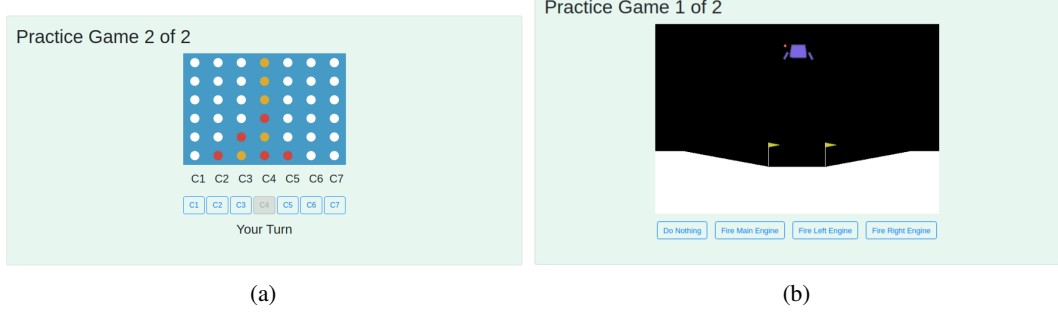

Figure 18: Visualizations of the game interface participants utilized during the *Practice*, *Pre-Test* and *Post-Test* stages for both Connect 4 (a) and Lunar Lander (b).

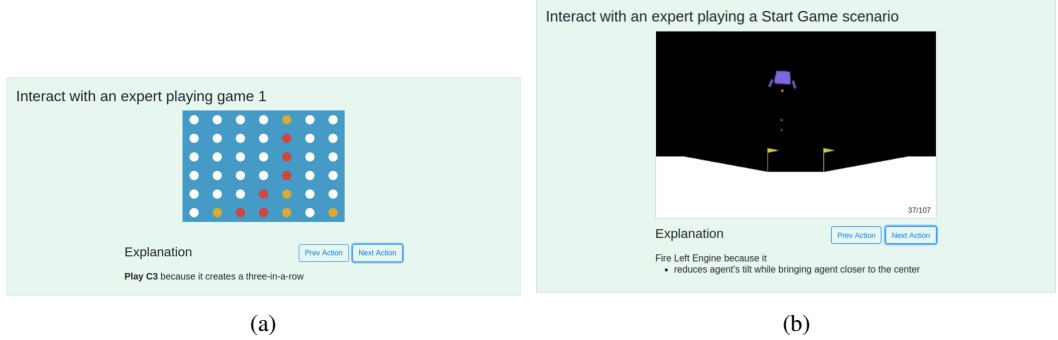

Figure 19: Visualizations of the *Explanation* stage for both (a) Connect 4 and (b) Lunar Lander.

introduction to the task was first presented to the users, and then the introduction to the domain was presented.

**Game-Playing Stages**  During the study, participants played various games during the *Practice*, *Pre-Test* and *Post-Test* stages. These games were in real-time; users clicked the action to play via a button, and saw the game interface update accordingly. Figure 18a(a) and Figure 18b(b) present visuals on how participants played games in each domain.

**Explanation Stage**  During the *Explanation* stage, participants interacted with an expert player (expert RL agent) and stepped through the expert player's actions while being exposed to explanations from a given study condition. The participants were not told that the expert player was an RL agent to limit the confounding effect of user biases towards AI agents in study. Figure 19a(a) and Figure 19b(b) provide an example of the explanation stage for Connect 4 and Lunar Lander. Specifically, in Figure 19a(a), a $\mathcal{E}_{C_i}$ is provided, and in Figure 19b(b), a $\mathcal{E}_{C_i \text{ w/ InF, TeG}}$ is provided.

**Survey**  At the end of the user study, participants were asked to fill out a short survey including demographic questions, as well as additional Likert questions to gauge user experiences with computers, the domain, as well as their perceived performance and explanation utility. Figure 20 provides a visual of what the survey looked liked in the domain of Connect 4. The survey for Lunar Lander participants was identical, except making an reference to Lunar Lander as opposed to Connect 4.

# F   Github Repo

https://anonymous.4open.science/r/S2E/README.md

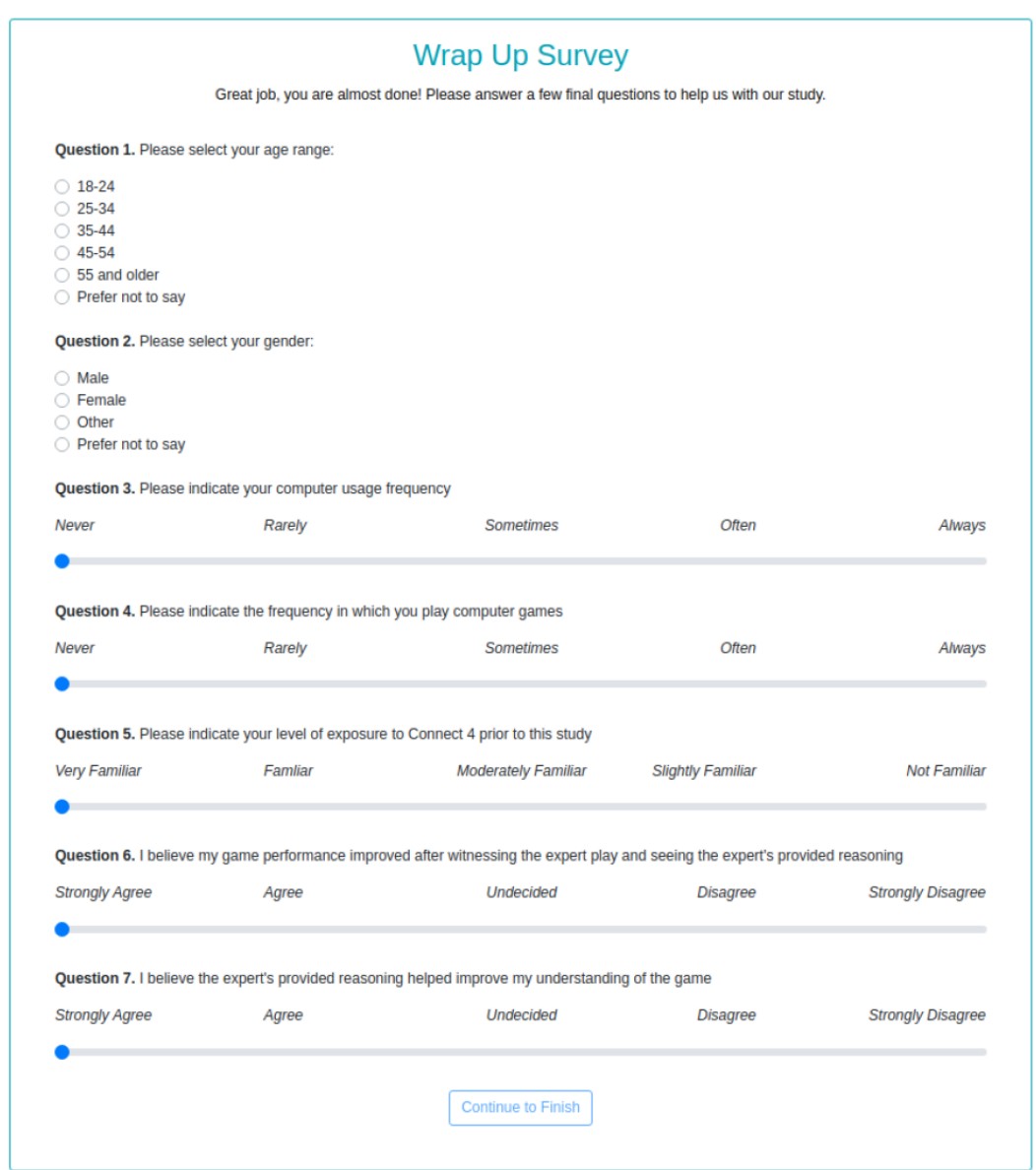

Figure 20: Visualization of the survey questions participants received at the end of the Connect 4 Study; questions were identical for the Lunar Lander study.

