# OpenReview forum: "State2Explanation: Concept-Based Explanations to Benefit Agent Learning and User Understanding"
_NeurIPS.cc/2023/Conference — NeurIPS 2023 poster_

### Official Review · Reviewer_GSRV · 2023-06-18

**Soundness:** 3 good
**Presentation:** 2 fair
**Contribution:** 2 fair
**Rating:** 4
**Confidence:** 4

**Summary:**

The paper proposes State2Explanation, a framework for training RL agents in such a way that both the human and agent benefit, the Protégé Effect as the authors state. The basic idea is to learn a joint embedding space with "temporal" concepts that actively helps the agent train better by shaping their rewards. At test time, these explanations are provided to humans to also help train them.

**Strengths:**

A strength of the paper is I feel the motivation. Explainability in RL is harder than supervised learning due to the temporal added component, so approaching this problem is well motivated.

In addition, the interplay between agent and human is also very exciting, I love the idea of the two helping and playing off each other. I feel this is a great way to impose human understandable concepts onto an RL agent's policy.

I also appreciate the rigour in the user study (although I need some clarifications in the rebuttal), it's clear to me the authors tried here.

**Weaknesses:**

What I always look for in a paper is a single nice idea, and I am not certain this paper really contains one, let me explain by iterating your claimed contributions.

Regarding the desiderata: (1) the idea that a 'concept should be grounded in human domain knowledge' is not a new idea I feel. The original T-CAV paper noted this by e.g. using zebra stripes to explain zebra classifications, rather than e.g. logit values. Simple saying the same thing is true in a sequential setting doesn't seem that original to me. I'm not aware of a great many people suggesting q-values are a good explanation. (2) The idea that a concept should relate to the task goal is an interesting one, and could be useful in certain context, but I feel it doesn't generalize well. For example, if I'm in a self-driving car going from a => b, and halfway through the journey the car brakes to avoid hitting a pedestrian, your framework seems to posit the explanation should be "I avoiding killing the human so I could still get to point b". I feel that's a questionable belief system here we are teaching the agent. (3) is a nice idea, but I don't feel it's particularly surprising, the idea that explanations should be robust and generalisable is well agreed upon.

The framework you propose does seem quite interesting to me, but reward feedback loops are not my area of expertise, so I defer to my colleges to evaluate the novelty of such a framework.

Section 5.2: I am slightly concerned that the explanations are not always matching in Figure 3, does that mean that this method will inevitably give "wrong" explanations? Also, I don't know what i, j, and k are in the figure? Now I personally believe it matters more that it just helps people in a user study, which you show it does, so that's great and I feel makes up for this.

Section 5.3: If I understand this correctly, your method doesn't seem to show much difference here at all, and in fact Fig 4a seems to show the baseline outperforming it?

User Evaluation: I appreciate the attention to detail here, but a lot of details are missing or (I feel) badly explained. I don't really understand how people's performance could get worse in group $E_{A}$, all they are being shown is the action as an explanation, so the idea their ATS would get worse seems extremely odd to me, and makes me feel I am misunderstanding something here. I tried to explore the appendix to understand what was happening, but didn't have much luck.

I think there's some citations that you might also like to be aware of regarding human friendly concepts in sequential settings.

Ji, Ying, Yu Wang, and Jien Kato. "Spatial-temporal Concept based Explanation of 3D ConvNets." Proceedings of the IEEE/CVF Conference on Computer Vision and Pattern Recognition. 2023.

Kenny, Eoin M., Mycal Tucker, and Julie Shah. "Towards interpretable deep reinforcement learning with human-friendly prototypes." The Eleventh International Conference on Learning Representations. 2023.

**Questions:**

1. During the training there's no feedback from the human right? If I understand, the human defines the concepts, and then these help the agent train better? (but Fig 4 seems to show this isn't the case?)
2. Whilst you used a CNN in connect 4, does this scale to harder Deep RL problems?
3. Is my understanding that Fig 3a shows the explanations are often "wrong" accurate?
4. What is i, j, and k in Fig 1?
5. How did you pick a sample size in the user study? Although it seems fairly rigorous, I don't see any power analysis, and you aren't clear what your attention checks are? It also seems strange that you had 66->60 and 98->90 in the two domains, a perfect balance for 15 in each group.

I might recommend using Prolific instead, it seems to have less issues compared to MTurk which is full of bots and LLMs.

I am happy to revise my review after the rebuttal.

**Limitations:**

This is ok

---

> ### Author Rebuttal · Authors · 2023-08-09
>
> Dear Reviewer GSRV, we thank you for your valuable feedback. Below we provide responses to weaknesses and questions:
>
> **W: single idea missing..**
>
> We disagree with this comment. To our knowledge, our framework S2E is the first unified framework that considers how concept-based explanations can provide a dual benefit to the RL agent and user, inspired by the Protege Effect. Specifically, S2E leverages a joint embedding model to retrieve concept-based explanations that both inform reward shaping to aid agent learning during training time, as well as improve user task performance at deployment time. Our work is important since S2E goes beyond showing the utility of explaining an agent’s behavior for user understanding and provides a joint benefit to the agent as well, all within a single framework.
>
> **W: desiderata..**
>
> Prior work in concept-based explanations for sequential decision making have had limited definitions of concepts using state preconditions, action costs or control logic [22,46 in paper]. However, our work posits that concepts in sequential decision making should be defined by higher-level properties as outlined by our desiderata. Our desiderata, inspired by concept-based explanations in classification tasks (like T-CAV), is a first step to providing a generalized definition of concepts in sequential decision making. We don’t claim that our desiderata is a complete set and will clarify this in Sec. 3. Future work includes extending our desiderata, including your valuable point about considering environmental uncertainties for defining concepts.
>
> **W: Sec. 5.2** + **Q3:**
>
> Fig. 3 shows the accuracy of the joint embedding models’ ability to correctly retrieve concept based explanations related to state-action pairs. Given the recall rates are not 100%, incorrect explanations are retrieved a fraction of the times as shown in Fig 3. However even with imperfect joint embedding models in S2E, results in Sec. 5 and Sec. 6 show S2E’s utility in a dual-benefit to agent learning rate and user task performance (see weakness response).
>
> **W: Sec. 5.3**
>
> In Fig. 4a, the current SoTA baseline agent that exists for Connect 4 is an agent not trained with reward shaping–referred to as “MuZero + No-RS”. Fig 4a shows that even with an imperfect joint embedding model, S2E is able to inform reward shaping and improve the agent’s learning rate compared to “MuZero + No-RS” in Connect 4. As mentioned in Sec. 5.3, our S2E informed reward shaping (“MuZero + S2E-RS”) improves the agent’s learning rate by 200 training steps compared to “MuZero + No-RS”. The “MuZero + E-RS” curve in Fig 4a shows the upper bound performance of S2E if our Connect 4’s joint embedding model was perfectly trained. We expect the small gap seen between “MuZero + E-RS”  and “MuZero + S2E-RS”, given from Fig. 3a that Connect 4’s joint embedding model is not 100% accurate. Similarly, in Fig. 4b and 4c we see that in Lunar Lander, our MuZero + S2E-RS can inform reward shaping compared to the existing soTA dense reward shaping (“MuZero + E-RS”). We will clarify this in Sec. 5.3.
>
> **W: User Eval..**
>
> This was an interesting finding; however, not a weakness of our method. The downward trend with “Action-Based” Explanations demonstrates possible negative effects of having to learn from only expert actions as feedback. We hypothesize that users provided with only expert actions caused users to project an incorrect reasoning to the experts’ actions which in return confused their understanding and led to worsened performance. We will discuss this finding in the appendix.
>
> **W: Citations** -- We appreciate you pointing out these; we will include in our related works.
>
> **Q1:**
>
> That is correct, during the RL agent’s training time there is no feedback from the human. To clarify, concepts are defined by domain experts and S2E learns a joint embedding model that maps concept based explanations to state-action pairs. The learned joint embedding model is then used to inform reward shaping during RL agent training and produce concept-based explanations at deployment to improve user task performance. See our response related to the weakness around Sec. 5.3 that our S2E can inform reward shaping and thereby improve agent learning rates compared to no-reward shaping (Connect 4), and provide comparable learning rates to existing dense-reward functions (Lunar Lander).
>
> **Q2:**
>
> CNN’s have been leveraged in existing Deep RL architectures, including the SoTA MuZero RL model architecture. We do not claim our exact CNN architecture will apply to all Deep RL problems, and modifications (such as additional layers in the joint embedding model) may be necessary for harder RL problems. Developing a single joint embedding model architecture generalizable across multiple complex RL domains was beyond the scope of this work and is mentioned in our limitations as future work.
>
> **Q4:**:
>
> The “i”, “j” and “k” in Fig. 1 describe three different concept-based explanation embeddings that exist in the learned joint embedding space. As noted in Sec. 4.3, the best ranked explanation embedding is retrieved from the learned joint embedding space similar to image-to-text-retrieval. We will qualify “i”, “j” and “k” in the Fig. 1 caption.
>
> **Q5:**
>
> It is good practice to have an equal number of study participants per study conditions to conduct fair statistical analysis comparisons. After filtering participants, we continued to recruit participants to ensure we had 15 participants per condition. We filtered participants by analyzing visualizations of the participants’ games. We determined a participant to show no effort if they only played a single action during the entire course of any of their pretest or posttest games. We did not conduct a power analysis in this particular study given that power analyses require making estimations about each study conditions’ mean or variance which we did not deem appropriate to assume. We will add these details to Appendix E.3.

---

> > ### Comment · Reviewer_GSRV · 2023-08-14
> > **Response**
> >
> > W: single idea missing..
> >
> > No problem, as I say that’s not my area of expertise, so I defer to the other reviewers to rate that aspect.
> >
> > ***
> >
> > W: desiderata..
> >
> > I understand, I agree with the sentiment of the desiderata, I just feel that it’s not really a significant contribution by itself, although a nice “aside” in the paper, something which is “backed up” by the subsequent experimental evidence.
> >
> > ***
> >
> > W: Sec. 5.2 + Q3:
> >
> > Ok I understand, thanks for the clarification here. Again, I don’t feel it’s a huge issue if explanations are occasionally wrong, it really depends on the downstream application we have in mind. E.g., on a Mars land rover I do feel they have to give certain guarantees, but if you’re just using the to improve task performance in another less sensitive way, I could imagine that’s ok. No pressure, but it could be worth mentioning.
> >
> > ***
> >
> > W: Sec. 5.3
> >
> > Ok thanks, for what it’s worth I personally struggled a bit to understand this, so it probably would be worth cleaning up the writing a bit.
> >
> > ***
> >
> > W: User Eval..
> >
> > If you could explain this in even more detail I would appreciate it, I still just really don't understand how this could happen. Even the experimental design?
> >
> > ***
> >
> > Q1:
> > Thank you, and just to clarify, these explanations are for actions right? Not a series of actions? That is, at each action taken, the user must interpret a separate explanation? Rather than say, one explanation for the next 20-40 actions?
> >
> > ***
> >
> > Q2:
> > Ok
> >
> > ***
> >
> > Q4::
> > Ok thanks.
> >
> > ***
> >
> > Q5:
> > It is good practice to have an equal number of study participants per study conditions to conduct fair statistical analysis comparisons.
> > 100% agree.
> >
> > After filtering participants, we continued to recruit participants to ensure we had 15 participants per condition.
> > But why 15? Why not 10? Why not 50? My concern is that it is well known you can “p-hack” your way to significance by simply increasing your sample size, that’s what a power analysis is for.
> >
> > We filtered participants by analyzing visualizations of the participants’ games. We determined a participant to show no effort if they only played a single action during the entire course of any of their pretest or posttest games.
> > Was there a large amount that played e.g. 2 actions per game? A single action seems a fairly arbitrary cutoff.
> >
> > We did not conduct a power analysis in this particular study given that power analyses require making estimations about each study conditions’ mean or variance which we did not deem appropriate to assume. We will add these details to Appendix E.3.
> > Ok. Usually you would observe similar studies done prior and base your assumptions on that. I know it is difficult, but user studies are not easy to do well.
> >
> > I’d appreciate if you could reply to the remaining concerns, thank you!

---

> > > ### Author Response · Authors · 2023-08-16
> > > **Response to GSRV - Part 1**
> > >
> > > Thank you for your follow up. We respond to your comments and concerns below. I've split my responses into two parts. This is part 1.
> > >
> > > **“Desiderata + single idea”:**  We also agree that our main contribution is the S2E framework. The desiderata is also a contribution given that we are expanding the existing definitions of concept-based explanations in sequential decision making problems.
> > >
> > > **“Sec 5.2 + Q2”:**  We agree and will add such details in the limitations section.
> > >
> > > **“Sec 5.3”:** We will improve the clarity of the current text in Sec. 5.3.
> > >
> > > **Q: “these explanations are for actions right...**
> > >
> > > In both user studies, during the “Explanation Stage”, the user sees an explanation per action taken by the RL agent in all study conditions. In the case of the concept based explanation with temporal grouping (TeG) condition, some explanations are grouped across identical sequential actions (see Sec. 4.3 and 6.2). However, the user still sees a single explanation per action, they are just made aware that such explanation holds true for “N” steps. For example, if the lunar lander agent “fires the right engine” 5 times consecutively. A concept-based explanation w/TeG would be: “Fire right engine for the next 5 steps to decrease agent tilt”, and the user would see such an explanation across the 5 steps. In comparison, a concept-based explanation without TeG would be “Fire right engine in this step to decrease agent tilt”, and the user would see such an explanation for each of the 5 steps.

---

> > > ### Author Response · Authors · 2023-08-16
> > > **Response to GSRV - Part 2**
> > >
> > > Thank you for your follow up. We respond to your comments and concerns below. I've split my responses into two parts. This is part 2.
> > >
> > > **User Eval: If you could explain this in even more detail..**
> > >
> > > In our previous response, we provide one hypothesis on the phenomenon of decreased user performance when exposed to “action-based” or “value-based” explanations. That is, these explanations may cause end-users to project wrong understandings to agent actions, and therefore their performances decrease in the post-tests. To study exactly why such decrease occurs requires performing more, in-depth qualitative user analyses such as conducting semi-formal interviews after the user studies to probe participants more about their experience and understand where such decreases may be coming from.
> > >
> > > **Experimental design:**
> > >
> > > We detail our user study in Appendix E.1 and provide visuals of the study in Appendix E.4. The user study consists of four stages in order:  “practice” stage, “pre-test stage”, “explanation stage” and “post-test” stage. Prior to the “practice stage”, participants are introduced to the task, and the game they will be playing. After the “post-test stage”, the user also answers a few survey questions in a questionnaire.
> > >
> > > The “practice stage” allows users to play 2 practice games–participants are told these are unscored games, and they are meant as practice to get users further familiarized with the game. The “pre-test stage” consists of playing 3 scored games. Users are told these games are an assessment and are scored. The “explanation stage” consists of the user interacting with a well-trained agent as the agent plays the game. The user is able to click through every action the agent plays in the game and depending on the study condition, the user sees an accompanied explanation type for each action played. In Connect 4, the user watches a Connect 4 RL agent play 3 games. In Lunar Lander, the user watches a Lunar Lander RL agent play 1 game. The “post-test stage” consists of playing 3 more scored games.
> > >
> > > As our analysis, we study how explanations provided in the “explanation stage” may help improve the user’s task performance. Therefore our metric is ATS, adjusted task score, is computed as a difference between the aggregated “post-test” task scores and the aggregated “pre-test” task scores. The single game’s task score is computed as an aggregation of user rewards received during the game (see Sec. 6).
> > >
> > > **Q5: But why 15?...sample size.,.power analysis"**
> > >
> > > The best way to address your concerns about our sample size per study condition having enough power is by demonstrating a priori power analysis as well as post-hoc power analysis. Please be aware that priori power analyses, as I mentioned earlier require setting assumptions. We will set the minimal assumption of effect size since we believe estimating group means are unrealistic when conducting such study for the first time. All power analyses are conducted in R using pwr.anova.test (prior power analysis) and power.anova.test (post-hoc power analyses). We will include these results in the appendix.
> > >
> > > The priori power analyses assumes an estimated power of 0.8 and estimated effect size of 0.4. This effect size is moderate, considering Cohen’s guidelines for effect size [1,2]. With these assumptions, we find N=15.89 for Connect 4 and N=13.09 for Lunar Lander. Our current study has N=15 for both studies, which falls within these estimated N values. We also perform a post-hoc power analysis given after performing the user studies we have the actual group means per study condition. The post-hoc power analyses show that our N=15 has a power of 0.99 and in Lunar Lander our N=15 has a power of 0.96. Thus, our statistical analyses reported have high statistical power. Overall, the above analyses show that N=15 is a reasonable number of participants to conduct for our user studies.
> > >
> > > 1. Cohen, J. (1988). Statistical power analysis for the behavioral sciences (2nd ed.). Hillside, NJ: Lawrence Erlbaum Associates.
> > > 2. Cohen, J. (1992). A power primer. Psychological Bulletin, 112, 155–159. doi:10.1037/0033-2909.112.1.155
> > >
> > > **Q5: A single action seems a fairly arbitrary cutoff...**
> > >
> > > Playing a single action was an obvious tell-tale that a participant was not performing the study with effort and was trying to finish as fast as possible. We visualized every participant’s actions in all games. There were only two distinctions we could fairly make. Participants who pressed a single button (single action) until the end of most games, and participants who did not press a single button and instead played actions that were more varied, showing effort. Trying to determine a more nuanced cutoff for filtering did not seem appropriate and could induce additional biases.

---

### Official Review · Reviewer_g9Rv · 2023-06-27

**Soundness:** 3 good
**Presentation:** 4 excellent
**Contribution:** 4 excellent
**Rating:** 7
**Confidence:** 4

**Summary:**

Inspired by the Protege effect, learning and developing explanations should provide a dual benefit, both to the readers of the explanations, and to the developers of the explanations. Based on this idea, the paper proposes State2Explanation, an algorithm to learn joint embeddings between state-action pairs and concept-based explanations. This allows for reward shaping, which benefits the explanation developer, while also providing explanations. These claims are validated for agents in two different reinforcement learning settings: Connect 4 and Lunar Lander.

**Strengths:**

1. Provides concrete desiderata for a concept is, including the need for generalizibility and its relationship to the task goal
2. Reward shaping through explanations is potentially novel, and explores use cases of explanations beyond explainability
3. Thorough evaluation investigates the impact of various additions to the model, making it clear what the impact of Information Filtering (InF) and Temporal Group (TeG) are
4. User study demonstrates real-world viability model through a human explainability lense


**Weaknesses:**

1. Concepts for domains are dependent on expert knowledge for state-action pairs, making it unclear how easy it would be to generalize beyond well-studied games. This is especially the case when state-actions pairs tend to get very large.
2. An additional baseline would make it clearer what the impact of each study condition was in Section 6. In particular, a baseline with no intervention/information would make it clear how much the rise in performance is due to the additional practice attained from the PreTest when completing the PostTest.
3. If concepts for Lunar Lander are derived from the existing domain reward function, then claiming that $M_{LL}$ "informs reward shaping comparable to expert-defined dense reward functions" seems to follow from the definition of the concepts rather than an indicator of the performance of the algorithm.


**Questions:**

1. For reward shaping, how is the amount of reward determined/amount the reward function is changed by?
2. In general, who would annotate the explanations, and what is the impact if there are multiple explanations for a single move/conflicting annotations?


**Limitations:**

The paper discusses limitations

---

> ### Author Rebuttal · Authors · 2023-08-09
>
> Dear Reviewer g9Rv, we thank you for your valuable feedback. Below we provide responses to weaknesses and questions:
>
> **W1: “Concepts for domains are dependent on expert knowledge for state-action pairs, making it unclear how easy it would be to generalize beyond well-studied games. This is especially the case when state-actions pairs tend to get very large.”**
>
> Thank you for highlighting this point. It is true that concepts related to state-action pairs should be derived from domain experts to ensure accurate concept representations. To scale to large state-action pair scenarios where collecting large amounts of expert annotated data may be unfeasible, our joint embedding model in S2E can be optimized for few shot learning. Note, a strength of leveraging a joint embedding model in our S2E is that our trained joint embedding model can also be used to retrieve accurate concepts-based explanations for unannotated state-action pairs. In this manner, when considering scenarios with large state-action pairs, we can leverage our joint embedding model to provide annotations as well. We will include this in the discussion section.
>
> **W2: “An additional baseline would make it clearer what the impact of each study condition was in Section 6. ….a baseline with no intervention/information would make it clear how much the rise in performance is due to the additional practice attained from the PreTest when completing the PostTest.”**
>
> It is true that an additional baseline where there is no information provided in the “Explanation stage” could have offered an additional layer of analyses when compared to our current study conditions.
>
> Importantly, we’d like to highlight that such additional baseline is **not necessary** to show “how much the rise in performance is due to the additional practice attained from the PreTest when completing the PostTest.” In our experimental analysis, we do currently take into consideration any learning effects obtained via the “PreTest” since our ATS metric measures the *difference* between the Pre-Test and Post-Test scores. That is, we account for participants’ initial performance on the “PreTest” and measure whether the intervention/explanations provided in the “Explanation stage” help to provide any *improvements* to their “Post Test” scores.
>
> **W3: “If concepts for Lunar Lander are derived from the existing domain reward function, then claiming that MLL "informs reward shaping comparable to expert-defined dense reward functions" seems to follow from the definition of the concepts rather than an indicator of the performance of the algorithm.”**
>
> We disagree with this comment. Concepts derived from an existing expert-defined reward function allow us to directly compare how well a joint embedding model trained using these concepts (M_{LL}) can inform reward shaping when compared to expert-defined reward shaping. In other words, having an existing dense reward function for Lunar Lander allows us to make a direct comparison of whether M_{LL} has learned good mappings between concepts and state-action pairs to effectively inform reward shaping.
>
> **Q1: For reward shaping, how is the amount of reward determined/amount the reward function is changed by?**
>
> Mentioned in Section 4.2 and 5.3, the actual amount of reward for the Lunar Lander domain is determined by the existing reward shaping function. Similarly for Connect 4, we perform a hyperparameter sweep to determine corresponding shaping values for each concept. In Appendix D.1 we have presented the shaping values for each concept for each domain. Note, we do not claim these are the most optimal shaping values, but the best that either currently exist (Lunar Lander) or were found via a hyperparameter sweep.
>
> **Q2: In general, who would annotate the explanations, and what is the impact if there are multiple explanations for a single move/conflicting annotations?**
>
> In general, concepts for any domain should be collected from experts within a domain. We will emphasize this in Section 4.1 and Section 5.1. As mentioned in Sec. 5.1,  the concepts in our work are identified by access to expert-domain knowledge within the fields.
>
> It is interesting to consider cases where “k” different concept-based explanations exist for a given state-action pair.  While in our work, we did not consider these scenarios, we believe there are at least two possible avenues to employ as solutions without drastically changing the S2E framework. One solution is to perform an inter-rater reliability to determine if there is agreement among experts on a single concept-based explanation out of the “k” explanations being more preferable or “agreed” upon, and using such single explanation when training the joint embedding model . A second solution is to allow the joint embedding model to see state-action pairs be associated with all “k” different concept-based explanations. During retrieval, the top ranked explanation can be retrieved, and given a well trained joint embedding model, the retrieved explanation will be one of the “k” possible correct explanations. These solutions are an opportunity for future work and we will include this discussion in an appendix that is related to Section 5.1.

---

> > ### Comment · Reviewer_g9Rv · 2023-08-10
> > **Response**
> >
> > Thank you for your clarifications. On the topic of the user evaluation my question was the following: Would the post-test agents not outperform the pre-test agents because the post-test agents see 3 more scored games, getting more practice? Therefore, would it not be more fair to have some set of agents practice -> explanation -> practice, and others do practice -> no explanation -> practice, then compare each?

---

> > > ### Author Response · Authors · 2023-08-10
> > > **Further Clarification on User Eval to Reviewer g9Rv**
> > >
> > > I appreciate your follow up. There seems to be an underlying misunderstanding that we did not catch earlier. In our user evaluation in Section 6, the RL agent’s policy in the user study is fixed and the agent is not learning in any way during our user study.
> > >
> > > We’d like to make this distinction very clear. The S2E framework provides a dual benefit of concept-based explanations to the RL agent and end-user, but at completely different stages. When we discuss the positive effects of concept-based explanations from S2E on agent learning, that occurs *during the agent’s training time*. The impact of S2E on agent learning is discussed in Section 5.  When we evaluate the benefit of concept-based explanations on end-users in Section 6, the RL agent has already been deployed, meaning the RL agent is already trained and its policy is fixed. Therefore, our user evaluation in Section 6 is focused on seeing how different explanation types of a fixed RL agent help participants improve their task performance and understanding. At the beginning of Section 6 we will reiterate this distinction to avoid future misunderstandings.
> > >
> > > With the above clarifications, we’d like to now respond directly to your questions.
> > >
> > > **Q: “Would the post-test agents not outperform the pre-test agents because the post-test agents see 3 more scored games, getting more practice? ..Would it not be more fair to have some set of agents practice -> explanation -> practice, and others do practice -> no explanation -> practice, then compare each?**
> > >
> > > There are no “post-test agents” or “pre-test agents” and there is no RL agent learning or practicing in the study. Instead, human participants play games in the “post-test” and “pre-test” stages. Furthermore, all participants play the same number of games, and no user receives more practice.
> > >
> > > Potential improvement in participant performance due to learning effects is accounted for in the metric. Please see our original rebuttal response on how our metric, ATS, takes user learning effects into consideration since the ATS metric measures the *difference* between the Pre-Test and Post-Test scores. In this manner, we account for participants’ initial performance on the “PreTest” and measure whether the explanations provided in the “Explanation stage” help to provide any *improvements* to their “Post Test” scores. Also, we’d like to point out that your question in the original review about having an additional baseline is still valid for human participants. That is, we could have had an additional baseline where human participants in the “Explanation Stage” received no feedback. However, as we mention in our original rebuttal response, our current study conditions are fair comparisons given that we account for learning effects as well as compare against relevant baselines utilized in prior work (Action-Based & Value Based Explanations).
> > >
> > > Please let us know if the above clarifies your understanding of our user study or if you have any other clarifying questions. Thank you!

---

> > > > ### Comment · Reviewer_g9Rv · 2023-08-11
> > > > **Response**
> > > >
> > > > Thank you for your response. I think there was some confusion over my use of the word "agent" as I was referring to human agents (AKA participants).
> > > >
> > > > If I understand correctly, do you guys measure the impact that having/not having the explanation has on the post-test score? Or do you measure the difference between post-test and pre-test score? If it's the latter, then how do you ensure that the participant seeing 3 more questions during post-test did not boost their post-test score over their pre-test score?

---

> > > > > ### Author Response · Authors · 2023-08-15
> > > > > **Response to g9Rv about additional "None" Study Condition + above question**
> > > > >
> > > > > Thank you for the follow up. Our ATS metric measures the difference between participant “Post-test” and “Pre-test” scores. We believe conducting the additional "None" study condition is the best way to analyze concerns about learning effects. Therefore, last week, we recruited participants from AMT and conducted the Lunar Lander and Connect 4 user studies with the “None” condition. Just as a recap, in the “None” condition, participants receive no explanations during the “Explanation Stage”. Each “None” condition includes 15 participants. Note, in total, we recruited 18 participants for Lunar Lander and 16 participants for Connect 4; however, we filtered 3 and 1 participants, respectively, based on our filtering criteria in Appendix E.
> > > > >
> > > > > Note, I am unable to attach the updated graphs from the paper since there's no option, it seems, to share a PDF and no links are allowed in our responses. However, the tables below summarize the post-test ATS scores. The values in the table are the values currently plotted in Figure 5 and Figure 6 in our paper. To both of these Figures, we will also add the “None” study condition results. Recall from the paper that the ATS metric adjusts the participants’ average  “Post-test” score by their average “Pre-test” score (see Appendix E.3). Therefore, the table shares the average adjusted “post-test” ATS per study condition.
> > > > >
> > > > > **Table Analysis:** The main takeaway to observe from the tables is the marginal difference between “None” and “Action-Based” for both Lunar Lander and Connect 4. These results show us that even without any information during the “Explanation Stage”, participants are not able to significantly improve their post-test scores from their pre-test scores.
> > > > >
> > > > > **Statistical Significance:** Our statistical analyses show that our S2E concept-based explanations significantly improve participant ATS compared to the “None” study condition. We perform a one-way ANOVA with a Tukey HSD post-hoc test for both Lunar Lander and Connect 4 as stated in our paper. In Connect 4, we find significant improvement on participant ATS with our S2E Concept-Based explanations in comparison to “None” (t(70) = 3.076, p<0.05).  In Lunar Lander, we find significant improvement on participant ATS with our S2E Concept-Based explanations that include TeG and InF abstractions in comparison to “None” (t(98) = 3.148, p <0.05). We will report these details in Section 5.
> > > > >
> > > > >
> > > > > **Connect 4 Table of average ATS score per study condition**
> > > > > | Study Condition | ATS: Post-Test|
> > > > > | -------- | ------- |
> > > > > | None | -0.038 |
> > > > > | Action-Based | -0.041  |
> > > > > | Value-Based | -0.104 |
> > > > > | S2E Concept-Based | 0.215 |
> > > > > | GT Concept-Based | 0.256 |
> > > > >
> > > > > **Lunar Lander Table of average ATS score per study condition**
> > > > > | Study Condition | ATS: Post-Test|
> > > > > | -------- | ------- |
> > > > > | None | -0.051 |
> > > > > | Action-Based | -0.049 |
> > > > > | Value-Based | -0.068 |
> > > > > | S2E Concept-Based | -0.043 |
> > > > > | S2E Concept-Based w/TeG | 0.100 |
> > > > > | S2E Concept-Based w/InF | 0.034 |
> > > > > | S2E Concept-Based w/TeG+InF | 0.156 |

---

### Official Review · Reviewer_ppHL · 2023-06-30

**Soundness:** 4 excellent
**Presentation:** 3 good
**Contribution:** 4 excellent
**Rating:** 7
**Confidence:** 4

**Summary:**

The paper proposes a framework to incorporate explanation concepts to sequential decision tasks. The framework can be applied both to the training of the agent by improving RL and to provide explanations to end-users during deployment. The framework is tested using two simple games, Connect 4 and Lunar Lander, where it shows it help improve training performance and user performance.

**Strengths:**

The main strengths of the paper are:

1. It provides an unified framework for explanations which can be used both for training and deploying.
2. The explanation framework is based on user-understandable concepts and terms.
3. The paper provides a sound theoretical and empirical analysis of the proposed framework.
4. The paper performs user testing and subsequent data analysis in a methodologically sound manner, rare in NeurIPs papers.


**Weaknesses:**

The main weaknesses of the paper are:

1. The paper condenses two much information in very little space, including impossible to read figures. It is very hard to read.
2. The paper fails to discuss how difficult will be to create frameworks for more complex scenarios where there are more and more complex concepts. For instance, consider a system making decisions about buying and selling stocks.
3. Because of 2, the ideas presented in the paper may never be applicable to real-world problems.
4. It provides a very brief description of how the concepts are actually created, stored, and verified.
5. The paper does not discuss situations where the explanation is wrong or inadequate for a situation (for instance, there is a better move), and how that impacts training and user performance.

**Questions:**

1. How do you expect the framework to perform when it is applied to more complex tasks and domains, both in terms of describing the actual concepts and on the performance during training?
2. How hard is to identify, create, store, and debug concepts?
3. What kind of wrong explanations it produces? How often? How that impacts training and user performance?

**Limitations:**

The paper does not do a good job of discussing how it can scale to real-world applications. By not doing so, it failed in identifying an important limitation of the work.

---

> ### Author Rebuttal · Authors · 2023-08-09
>
> Dear Reviewer ppHL, we thank you for your valuable feedback. Below we provide responses to weaknesses and questions:
>
> **W1:** Thank you for pointing out--we will increase our image sizes.
>
> **W2 & W3:**
>
> We believe that our S2E framework will be applicable to more complex scenarios; however, two components within the framework may need modifications for scalability. We will add the discussion of these two components and their importance in the discussion section.
>
> First, with an increased number of concepts (i.e. Chess) coupled with some concepts likely to occur more rarely (i.e. castling in Chess), we hypothesize that leveraging more complex joint-embedding model architectures that favor few-shot learning may be necessary. Second, with more complex domains there may be a greater need for providing abstracted concept-based explanations to end-users. S2E currently includes temporal grouping and information filtering, which we observed to be important for improved user task performance in Lunar Lander. We believe such abstraction methods will continue to be important in other complex domains.
>
> Overall, our evaluations in Lunar Lander and Connect 4 demonstrate the efficacy of the S2E framework. We believe its modular design (a joint embedding model learning decoupled from an explanation abstraction method) allows for implementing and validating extensions to both components important in providing a dual-benefit to agent and end-user.
>
> **Q1:**
>
> As mentioned in response to the weakness above, we expect our S2E framework to be applicable in more complex tasks and domains. We expect some modifications needed to the joint-embedding model architecture to provide accurate retrievals of concepts to state-action pairs. If the joint-embedding model can retrieve concepts with high Recall@1, then based on our current findings, S2E can provide accurate concept-based explanations that both benefit end-user understanding at deployment as well as improve agent learning during training time.
>
> **W4:**
>
> We will add the following to Section 5.1 as more details.In our work, each state is associated with relevant concepts by objective, mathematical representations of each concept (i.e. function for position over time, rule for existence of a three-in-a-row, etc). Therefore we verify that concepts are accurately paired by ensuring the mathematical rule for a concept to be paired to a state-action pair is met.
>
> In the appendix we will also provide a more detailed guideline (with examples) on how we derive and verify concepts mathematically, and discuss how to derive and verify concepts that are collected via human annotation or commentary.
>
> **Q2:**
>
> In our work, we leverage prior expert domain knowledge in Lunar Lander and Connect 4 to derive concepts important to the games via mathematical rules. In many real-world applications, it may be infeasible to leverage mathematical rules for concept derivation. Instead, concepts can be collected via crowdsourcing, such as in [20 from paper], or obtained via “think-aloud” procedures mentioned in [14 from paper]. Note, when concepts are collected via domain expert labeling, such labels should be verified via inter-rater reliability tests to ensure consistency among concept representations. These details will be added to the Appendix section.
>
> **W5:** + **Q3:"..What kind of wrong explanations..How often?...”**
>
> Figure 3a and 3b show the accuracy of our trained joint embedding models in retrieving concept-based explanations within Lunar Lander and Connect 4. Specifically, Fig. 3b quantifies a breakdown of the incorrect concept-based explanation retrievals in Connect 4 on the joint embedding model’s test set. The learned joint embedding model for Lunar Lander has a near-perfect recall@1(99.9). In the Appendix, we will include example scenarios where the retrieved concept-based explanations are incorrect.
>
> **Q3:“..How [wrong explanations] impacts training and user performance?..”**
>
> Incorrect retrievals of concepts-based explanations for a given state-action pair can negatively impact agent training in S2E by incorrectly providing shaping rewards to the agent and in return impacting learned agent policy. However, our results in Fig. 4a and 4b show that the percentage of incorrect retrievals from our joint embedding models do not significantly impact the RL agent’s learning rate. In Fig. 4a when studying Connect 4, the reward shaping informed via S2E (MuZero + S2E-RS) results in a slightly lower learning rate than the upper bound expert-defined reward shaping (MuZero + E-RS). However, MuZero + S2E-RS still improves the agent’s learning rate by ~200 training steps compared to the SoTA baseline agent (MuZero + no-RS). Similarly, in Fig. 4b and 4c, we see that MuZero + S2E-RS performs similarly to the SoTA MuZero + E-RS, given that there exists expert-defined reward shaping for Lunar Lander. We will clarify this in Sec. 5.3.
>
> With respect to user performance, concept-based explanations provided to the end-users during the “Explanation” stage are retrieved from our learned joint-embedding models. However, the number of incorrect retrievals was not significantly detrimental to user performance. As shown in Fig. 5, while the ground truth concept-based explanation condition had slightly greater ATS improvement, there is no significant difference in ATS improvements between the ground truth concept-based condition and its S2E counterpart. For Lunar Lander, we did not experiment with a ground truth concept-based condition given that our Lunar Lander joint-embedding model had near-perfect Recall@1 (99.9). We hypothesize that if we did perform such a comparison, we’d see no significant differences given the near-perfect Recall@1. We will clarify this in Sec. 6.1 and 6.2.

---

> > ### Comment · Reviewer_ppHL · 2023-08-14
> >
> > Thanks to the authors for the information in the rebuttal.
> >
> > I confirm I have read the rebuttals provided by the authors.

---

### Official Review · Reviewer_smxt · 2023-07-04

**Soundness:** 3 good
**Presentation:** 3 good
**Contribution:** 3 good
**Rating:** 6
**Confidence:** 4

**Summary:**

The authors propose a unified framework called State2Explanation (S2E) that combines learning a joint embedding model between state-action pairs and concept-based explanations. The authors draw inspiration from the Protégé Effect, which suggests that explaining knowledge reinforces self-learning. They propose that concept-based explanations can benefit both the RL agent and the end-user by improving the agent's learning rate and the end-user's understanding of the agent's decision making. The S2E framework is designed to inform reward shaping during an agent's training and provide explanations to end-users at deployment for improved task performance.Results on Connect 4 and Lunar Lander demonstrate the success of S2E in providing a dual benefit.

**Strengths:**

The paper introduces a novel framework S2E. The authors suggest that explaining knowledge to the agent can improve its learning rate, while providing explanations to end-users can enhance their understanding of the agent's decision making. The framework is supposed to be useful in providing explanations understandable in various applications.

**Weaknesses:**

There are no significant weaknesses from my point of view, while there are some improvements can be made. Although the paper mentions experimental validations in the Connect 4 and Lunar Lander domains, the proposed method should be tested on more complex tasks to test its effectness. The authors should also consider how concepts can be better selected beyond expert-defined thresholding.

**Questions:**

- To be specific, how concept candidates are selected when performing InF?
- How thresholds are determined via qualitative methods? Is there any results showing that different concepts selected will influence the final performance?

**Limitations:**

See weaknesses and questions.

---

> ### Author Rebuttal · Authors · 2023-08-09
>
> Dear Reviewer smxt, we thank you for your valuable feedback. Below we provide responses to weaknesses and questions:
>
> **W: “..the proposed method should be tested on more complex tasks to test its effectiveness. The authors should also consider how concepts can be better selected beyond expert-defined thresholding.”**
>
> We agree that more investigation of the applicability of S2E should be performed in other complex domains.  However, we believe insights from Connect 4 and Lunar Lander are themselves valuable, as both domains are challenging and complex in their own ways. We chose Connect 4 given its large state space (~4 trillion unique states), and Lunar Lander for its complex continuous state space with actions sampled at high frequency, properties that make both domains challenging in RL as well as for providing explanations.
>
> We believe S2E can be applicable in other settings, such as domains with more complex trajectory optimization (robotics) or complex concept representations (i,e. Go, Chess) with small modifications made to the joint embedding architecture to improve concept-explanation to state-action pair mappings. We agree that future work should investigate other automated methods for concept-filtering beyond expert-defined thresholding. We will add this to our limitations section.
>
> Overall, we would like to highlight that given, to our knowledge, that this is the first work that explores the notion of a dual-benefit of concept-explanations to both the agent and end-user, we believe that our proposed S2E framework and its success in providing a dual benefit in Lunar Lander and Connect 4 is an important contribution towards understanding how concept-based explanations can be utilized to provide a dual benefit in other RL domains.
>
> **Q1:To be specific, how concept candidates are selected when performing InF?**
>
> In InF, the thresholds are defined through a qualitative analysis of the agent’s state values when rolling out the agent’s policy. Note, the state values that are analyzed are directly mapped to a concept(s). For example, in Lunar Lander we observe the agent’s “x-position” and “tilt” state values which correspond to concepts “position” and “tilt”. These details are provided in Appendix A.
> For more details, we will add the following details to Appendix A: The derived thresholds signify the positive and negative turning points in the agent’s ability to reach G. In other words, these thresholds are expert-defined upper and lower bounds on the agent’s state values that denote the agent’s ability to succeed or fail in its goal. In our InF method, these thresholds are not mathematically derived, but are derived from RL-expert analysis. Specifically, an RL expert visualizes multiple policy rollouts while analyzing the different state values over time to manually determine upper and lower bounds (turning points) that influence the agent’s ability to reach G. In our discussion section we will include that future work involves automating the InF method.
>
>
> **Q2:How thresholds are determined via qualitative methods? Is there any results showing that different concepts selected will influence the final performance?**
>
> As mentioned in response to Q1, the thresholds are manually derived from RL-expert analysis. Specifically, an RL expert visualizes multiple policy rollouts while analyzing the different state values over time to determine approximate upper and lower bounds (turning points) that influence the agent’s ability to reach G. It is true that different thresholds can result in different abstracted, concept-based explanations.
>
> In response to the rebuttal, we performed an additional study to analyze the sensitivity of our current chosen thresholds for the Lunar Lander domain. Check the PDF in the global review for the graphs. We will add these graphs to Appendix A as additional analyses. All graphs show what fraction of concepts are filtered (y-axis)  as the threshold values change (x-axis). When looking the first graph, “X-position: Threshold Experimentation”, we see our chosen value is within the elbow of the curve, denoting that the rate of filtration of the “position” concepts slows down after 0.15. Similarly, in the “tilt” experimentation graphs (2nd and 3rd graphs), we see the lower and upper bound of the tilt thresholds to also be within the “elbow” in each curve. Note, in the upper bound threshold experimentation, the lower bound value is fixed, and vice versa.  To study the impact of these threshold values, additional user studies need to be conducted on the utility of the different abstracted concept-based explanations that result from varying the experimental threshold values.  However, we consider such analyses beyond the scope of our work, and one to further explore when improving the information filtering submodule within our S2E framework.

---

> > ### Comment · Reviewer_smxt · 2023-08-17
> > **Thanks**
> >
> > Thanks to the authors for the information in the rebuttal. I have read the author's response and fellow reviewer's feedback. I believe that the authors have addressed most of my concerns.

---

### Author Rebuttal · Authors · 2023-08-09

We thank all our reviewers for their detailed comments. Firstly, we are encouraged that reviewers saw the importance of our framework S2E in providing a dual benefit to the end-user as well as RL agent. Reviewer ppHL found our work provided “sound theoretical and empirical analysis”, and reviewer g9Rv saw strength in our “concrete desiderata for a concept”. Reviewer smxt believed in the novelty of our work, and reviewer GSRV agreed with the important need for a framework like S2E, and loved the “idea of the two [agent and user] playing off each other”. We are also glad that reviewers saw value in our analyses, in particular that our user studies demonstrated “real-world viability” (g9Rv), was “methodologically sound” (ppHL) and its “rigor” was appreciated (GSRV).

We found all feedback to be constructive and informative. We respond to each reviewer’s questions and weaknesses below. We believe that our responses should clarify and address reviewer concerns. If additional details or clarifications are needed, we are happy to provide them.

**Reviewer smxt**: please see attached graphs that support our response to your Q2.

---

### Decision · Program_Chairs · 2023-09-21

**Decision:**

Accept (poster)

**Comment:**

The work introduces a framework for using explanations to guide representation learning. They demonstrate the effectiveness of their approach on two simple domains (Connect 4 and Lunar Lander) which have descriptions that correspond to actions, states, and basic strategy.  Authors were happy with the work and its analysis.  The primary shared concern is generality to more complex and realistic settings.

The final draft can be strengthened by addressing several natural concerns raised during the reviewing.  In particular, the amount of human effort, how this will change with more complex domains, and how the approach handles incorrect explanations.